# POSITION-AWARE MODELING FOR NEXT-TOKEN PREDICTION

## ABSTRACT

Next-token prediction (NTP) serves as the dominant training paradigm for large language models (LLMs), enabling strong autoregressive (AR) generation capabilities. Despite its success, models trained with vanilla NTP often exhibit counterintuitive failure patterns, such as the reversal curse, factorization curse, and sensitivity to knowledge position. These failures stem from the lack of permutation invariance in LLMs, which arises from the fixed left-to-right token order used during teacher-forcing supervision. To address this issue, we introduce a position-aware training framework that enables AR models to learn from all possible permutations of the sequence. We begin by introducing a position-aware embedding that enables LLMs to predict the next token not only based on the preceding context, but also by incorporating its position within the sequence. This embedding is integrated into LLMs through two complementary approaches: (1) Content-Position Coupling (CPC), which injects the embedding directly into the input embedding via element-wise addition, without altering the model architecture; and (2) Content-Position Decoupling (CPD), which adds modular position-aware blocks with a cross-attention mechanism on top of AR models. In this mechanism, the position-aware embedding serves as the query, while the hidden states from the final layer of the AR model serve as the key and value. Experiments across three representative tasks demonstrate that our framework consistently improves performance over strong baselines, while maintaining architectural simplicity and convergence efficiency. Codes are available at `https://anonymous.4open.science/r/CPC-CPD`.

## 1 INTRODUCTION

Next-token prediction (NTP) is the primary pre-training objective for large language models (LLMs) (OpenAI, 2023; Touvron et al., 2023a). LLMs can effectively learn co-occurrence patterns among tokens by optimizing the autoregressive (AR) maximum likelihood estimation objective on large text corpora (Zhang et al., 2024), thereby facilitating the transfer of learned knowledge to diverse applications, ranging from text generation to complex question answering and reasoning (Petroni et al., 2019; Hendrycks et al., 2020). NTP commonly integrates the teacher forcing mechanism (Williams & Zipser, 1989) during the training phase and employs AR at inference time (Bachmann & Nagarajan, 2024). Owing to its significant advantages-notably in training efficiency (Gloeckle et al., 2024; Li et al., 2024), gradient stability (Chen et al., 2024), and amenability to parallel computation (Li et al., 2021; Rasley et al., 2020), NTP has been established as a cornerstone in the pre-training of mainstream LLMs (OpenAI, 2023; Touvron et al., 2023a; Liu et al., 2024a; Jiang et al., 2024a; Bai et al., 2023).

Despite its long list of achievements, existing research has discovered that models trained via vanilla NTP can surprisingly exhibit counterintuitive failure patterns (Berglund et al., 2024; Lin et al., 2024; Lv et al., 2024; Bachmann & Nagarajan, 2024; Kitouni et al., 2024; Allen-Zhu & Li, 2024; Saito et al., 2025). For instance, they may suffer from (1) the **reversal curse** (Berglund et al., 2024; Lin et al., 2024; Lv et al., 2024), where learned factual associations (*e.g.*, "A is B") fail to generalize to their inverse form (*e.g.*, "B is A"); (2) the **factorization curse** (Kitouni et al., 2024), which arises when the model, trained on a specific decomposition of the token sequence (*e.g.*, left-to-right), fails to represent the same joint distribution under alternative factorizations; and (3) the **knowledge position sensitivity** (Allen-Zhu & Li, 2024; Saito et al., 2025), where factual information encoded during training is only reliably accessible when it appears in early positions of

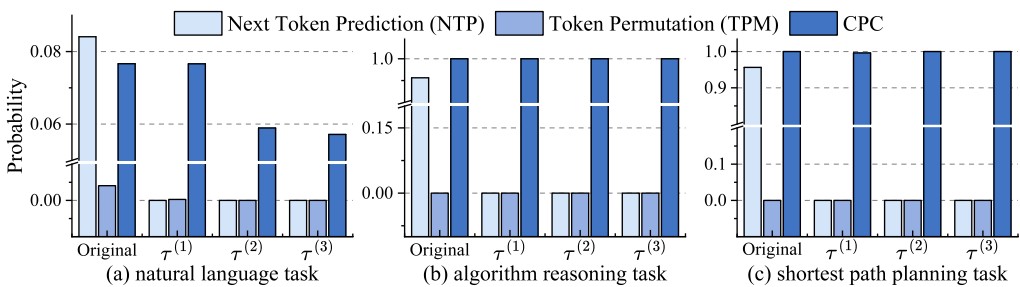

Figure 1: Joint probability across different permutations on the same sample under three task types. Our method maintains nearly consistent joint probability across different permutations, while both NTP and TPM fail to achieve probability invariance. $\tau^{(i)}$ denotes a specific permuted token order. For more detailed experimental settings and more examples, see Appendix D.1.

the training document, while knowledge located later is often unrecoverable during inference, even with elaborately designed prompting. These failure patterns reveal a shared deficiency: **the lack of permutation invariance in vanilla NTP**. Specifically, models trained only on the fixed left-to-right sequence fail to maintain consistent joint probability distributions across different permutations of the same content. As an example, when exposed to the sentence "Paris is the capital of France" during training, the model is optimized to maximize the joint probability of that particular token ordering. Conversely, the semantically equivalent permutation "The capital of Paris is France" receives a probability approaching zero under the learned distribution. As illustrated in Figure 1, vanilla NTP assigns high probability only to the original sequence during training, while probabilities for other permutations (*i.e.*, $\tau^{(\cdot)}$) drop nearly to zero. This deficiency hinders the model's ability to generalize to alternative token orders, thereby impairing its performance across a wide range of tasks, including natural language understanding, algorithmic reasoning, and planning.

Existing research that mitigates these pitfalls can be divided into two major directions. **Data-centric** strategies include data rewriting and token permutation (TPM) (Golovneva et al., 2025; Guo et al., 2024) to encourage model learning under diverse token factorizations, and structural reorganization of training data to break the inherent left-to-right learning pattern of NTP, *e.g.*, by exposing future tokens in advance to models (Thankaraj et al., 2025). **Model-level** work equips AR models with bidirectional attention mechanisms to better capture global contextual dependencies (Lv et al., 2024).

However, there are two primary challenges: (1) For data-centric methods, data rewriting typically relies on advanced LLMs (*e.g.*, GPT-5), which inevitably introduce hallucinations. Moreover, TPM under vanilla NTP often causes **different target tokens to share identical prefix sequences**, creating supervised label conflicts that undermine training stability.[1] As shown in Figure 1, similar to NTP, TPM struggles to assign consistent probabilities across various permutations, even after sufficient training, and **especially in planning and algorithm reasoning, it underperforms compared to vanilla NTP**. (2) For methods that involve modifying the model architecture or training objective, making them difficult to generalize across different backbone architectures. Moreover, applying such architecture or objective changes directly to pre-trained LLMs creates a significant mismatch between the fine-tuned and the original model, potentially degrading acquired abilities.

In this work, we leverage token permutation to expose the model to diverse token orderings, thereby encouraging the model to learn position-agnostic representations and ultimately achieve permutation invariance. To address the inevitable issue of conflicting supervision signals introduced by TPM, where different ground truths are associated with the same prefix, we augment the vanilla NTP objective with position-aware modeling, explicitly encoding the positional information of the target token. Specifically, we introduce a single learnable base positional embedding and then rotate it to arbitrary positions via rotary position embedding (RoPE) to generate the target position-aware embeddings. By incorporating these embeddings, the model learns to predict the next token not only based on the preceding content but also on its position within the sequence, thereby mitigating conflicting supervision signals of token permutations. Concretely, we introduce two complementary approaches to integrate target position-aware embeddings: (1) **Minimal modification, Content-Position Coupling (CPC)**: This approach preserves the original AR architecture and directly integrates the target position-aware

---

[1]A detailed discussion of this issue is provided in Appendix E.1.

embeddings (*i.e.*, position) into input embeddings (*i.e.*, content) of permuted sequences through element-wise addition, introducing only minor modifications to the input layer of models. As shown in Figure 1, CPC can maintain almost the same joint probability for different permutations. (2) **Incremental module, Content-Position Decoupling (CPD)**: While CPC provides a lightweight solution, its direct integration of target position-aware embedding and input embeddings may degrade the capabilities acquired during pre-training. To address this, we further propose CPD, which explicitly decouples content and positional information by incorporating incremental position-aware blocks on top of the pre-trained AR models. These modular blocks employ cross-attention mechanisms, where target position-aware embeddings serve as queries and the hidden states of the pre-trained AR models serve as keys and values, without modifying the original input representation. Crucially, CPD requires no architecture changes and can be integrated into any pre-trained AR models as a learnable module, enabling position-aware adaptation without compromising the model's original capabilities. We summarize our contributions below.

- We reveal that seemingly disparate failure patterns in LLMs actually stem from a single fundamental limitation: the lack of permutation invariance under vanilla NTP training, which particularly impairs models' planning and algorithm reasoning capabilities.

- We propose the position-aware modeling framework that enables models to predict the next token not only based on the preceding content, but also by incorporating its position within the sequence, thereby achieving permutation invariance.

- Extensive experiments demonstrate that our proposed methods significantly enhance model robustness to token order, enabling smaller LMs to outperform larger backbone models. Notably, CPD achieves a balance between mitigating NTP failures and preserving original capabilities.

## 2 RETHINKING FAILURE PATTERNS IN NTP

### 2.1 PRELIMINARIES

Consider a sequence $s = (p, r)$, where $p = (p_1, p_2, \ldots, p_{|p|})$ denotes the prompt with position index $\tau^{(1)} = \{0, 1, \cdots, |p| - 1\}$ and $r = (r_1, r_2, \ldots, r_{|r|})$ denotes the response with position index $\tau^{(2)} = \{0, 1, \cdots, |r| - 1\}$. Each token $p$ and $r$ is drawn from a fixed-size vocabulary $\mathcal{V}$. For each position $t_{th}$ in the sequence $s$, let $s_{<t}$ denote the subsequence consisting of the first $t - 1$ tokens and $s_t$ denote the token at position $t$. Suppose we have a NTP language model $P_\theta$ parameterized by $\theta$, such that $P_\theta(s_t \mid s_{<t})$ denotes the probability that the model assigns to the $t_{th}$ token $s_t$, conditioned on the preceding sequence $s_{<t}$. For the given sequence $s$, the joint probability is axiomatically defined analogous to the chain rule of probability:

$$P_\theta(r \mid p) = \prod_{t=1}^{|r|} P_\theta\left(r_t \mid p, r_{<t};\ \tau^{(1)}, \tau^{(2)}_{<t}\right) \qquad (1)$$

Here, explicitly displaying the position index $(\tau^{(1)}, \tau^{(2)}_{<t})$ in Eq. 1 does not imply that it is tokenized as part of the input sequence. Instead, it serves to instruct the model's internal positional encoding mechanism in assigning positional information to each token.

**Training-time next-token prediction via teacher-forcing**   To train the above NTP model, mainstream LLMs adopt teacher forcing to maximize the log-probability sum of the next token, where the model is trained to predict each token $r_t$ using the ground-truth $r_{<t}$ as input. The teacher-forcing objective $\mathcal{J}_{\text{teacher-forcing}}(\theta)$ on dataset $\mathcal{D}$ can be formulated as follows:

$$\mathcal{J}_{\text{teacher-forcing}}(\theta) = \mathbb{E}_{(p,r)\sim\mathcal{D}}\left[\log P_\theta(r \mid p)\right] = \mathbb{E}_\mathcal{D}\left[\sum_{t=1}^{|r|} \log P_\theta\left(r_t \mid p, r_{<t};\ \tau^{(1)}, \tau^{(2)}_{<t}\right)\right] \qquad (2)$$

**Inference-time next-token prediction via autoregression**   During inference, the model is conditioned on a given prompt $p$ and generates response tokens $\hat{r}$ by sequentially sampling from the learned distribution $P_\theta$. Specifically, for each step $t$, the model samples a token $\hat{r}_t \sim P_\theta(\cdot \mid p, \hat{r}_{<t})$, where $\hat{r}_{<t}$ represents the previously generated tokens. The sampled token $\hat{r}_t$ is appended to the existing context and then provided as input to the model for the next prediction. A full sequence is formed by this autoregressive generation process continuing for $|r|$ steps.

## 2.2 MITIGATING FAILURE PATTERNS IN NTP

Building on the insight by Kitouni et al. (2024) that consistency across token factorizations improves knowledge retrieval, we generalize this goal to a broader perspective. We argue that the observed failure patterns in NTP, namely the reversal curse, factorization curse, and knowledge position sensitivity, reflect a shared underlying limitation in vanilla NTP: the lack of permutation invariance.

**Permutation invariance** Let $\tau^{(i,n)} \in S_n$ be the $i_{th}$ sampled permutations, where $S_n$ is the set of all $n!$ permutation of the indices $\{1, 2, \ldots, n\}$. Thus, $\tau^{(i,n)} = \{\tau_1^{(i,n)}, \tau_2^{(i,n)}, \ldots, \tau_n^{(i,n)}\}$. Applying permutation $\tau^{(i,|\boldsymbol{p}|)}$ and $\tau^{(i,|\boldsymbol{r}|)}$ reorders sequence tokens accordingly, yielding permuted prefixes $\boldsymbol{p}_{\tau^{(i,|\boldsymbol{p}|)}}$ and responses $\boldsymbol{r}_{\tau^{(i,|\boldsymbol{r}|)}}$. Then, for two sampled permutations $\tau^{(i,|\boldsymbol{p}|)} \in S_{|\boldsymbol{p}|}, \tau^{(i,|\boldsymbol{r}|)} \in S_{|\boldsymbol{r}|}$, the permutation invariance expect model $P_\theta$ could assign approximately consistent joint probability across different permutations of the input. With an abuse of notation, let $\boldsymbol{p}_{\tau^{(\boldsymbol{p})}}$ and $\boldsymbol{r}_{\tau^{(\boldsymbol{r})}}$ denote a permutation of prompt and response, respectively. Permutation invariance can be formulated as:

$$\prod_{t=1}^{|r|} P_\theta \left( r_{\tau_t^{(\boldsymbol{r})}} \mid \boldsymbol{p}_{\tau^{(\boldsymbol{p})}}, \boldsymbol{r}_{<\tau_t^{(\boldsymbol{r})}};\ \tau^{(\boldsymbol{p})}, \tau_{<t}^{(\boldsymbol{r})} \right) \approx \prod_{t=1}^{|r|} P_\theta \left( r_{\tau_t^{(2)}} \mid \boldsymbol{p}_{\tau^{(1)}}, \boldsymbol{r}_{<\tau_t^{(2)}};\ \tau^{(1)}, \tau_{<i}^{(2)} \right) \quad (3)$$

where $\tau^{(1)}$ and $\tau^{(2)}$ respectively denote the token order of the prompt and the response in natural language during training. Importantly, permutation invariance does not mean models assign identical joint probabilities to any permutation. Instead, it refers to **semantically equivalent permutations** in which, when the token order is permuted, the model's internal positional encoding is correspondingly adjusted so that the semantic remains consistent with the underlying content.

To achieve permutation invariance in Eq. 3, the straightforward strategy is to permute the training data sufficiently and then optimize vanilla NTP, which can be formulated as follows:

$$\mathcal{L}_\theta = \mathbb{E}_{(\boldsymbol{p},\boldsymbol{r})\sim\mathcal{D}} \mathbb{E}_{\tau^{(p)}\sim\mathcal{S}_{|\boldsymbol{p}|}, \tau^{(r)}\sim\mathcal{S}_{|\boldsymbol{r}|}} \left[ \sum_{t=1}^{|r|} \log P_\theta \left( r_{\tau_t^{(\boldsymbol{r})}} \mid \boldsymbol{p}_{\tau^{(\boldsymbol{p})}}, \boldsymbol{r}_{<\tau_t^{(r)}};\ \tau^{(\boldsymbol{p})}, \tau_{<t}^{(\boldsymbol{r})} \right) \right] \quad (4)$$

Although Eq. 4 ensures that the positional information is adjusted accordingly after permutation, this operation inherently introduces a fundamental conflict: **given the same prefix, the model is required to optimize for different next-token targets, which results in conflicting supervision signals.** Moreover, prior studies (Kitouni et al., 2024) have demonstrated that the masked language modeling (MLM) objective is effective in alleviating both the reversal curse and the factorization curse. It randomly masks tokens at arbitrary positions and predicts them using bidirectional context, allowing the model to learn representations that are inherently robust to various token orders. However, it has not been incorporated into the prevailing pre-training paradigms of existing LLMs, as its implementation often requires modifications to the internal attention mechanism (Lv et al., 2024) or complete model re-training. Such interventions may conflict with the intrinsic AR pre-training objective or impose substantial computational overhead. To achieve the permutation invariance within the pre-trained AR models, it is desirable to combine the AR structure of NTP with the positional flexibility of MLM, *i.e.*, enabling the model to learn from the same training sample under arbitrary token permutations during the training process. This requires explicitly identifying which token is to be predicted under each permutated context.

## 3 METHODOLOGY

Considering the conflicting supervision signals brought by token permutations, we propose a target position-aware training framework, introducing target position information into NTP. By extending Eq. 1, we perform NTP conditioned not only on the content and positions of preceding tokens, but also on the position of the target token. Specifically, the probability of target token $s_{\tau_t}$ can be formulated as follows[2]:

$$P_\theta(s_{\tau_t} \mid \boldsymbol{s}_{<\tau_t}) = P_\theta \left( s_{\tau_t} \mid \left\{ \sigma(\text{Embed}(s_{\tau_j}), \text{Pos\_Embed}(\tau_j), \boldsymbol{z}_{\tau_{j+1}}) \right\}_{j<t} \right) \quad (5)$$

where $\boldsymbol{z}_{\tau_{j+1}}$ $(j+1 \leq t)$ is the target position-aware embedding of position $j+1$, $\text{Embed}(s_{\tau_j})$ denotes the embedding of the content $s_{\tau_j}$, and the position encoding $\text{Pos\_Embed}(\cdot)$ can be either the absolute

---

[2]Without loss of generality, NTP is not limited to prefix-prompted generation, as it can likewise be trained directly on prompt tokens.

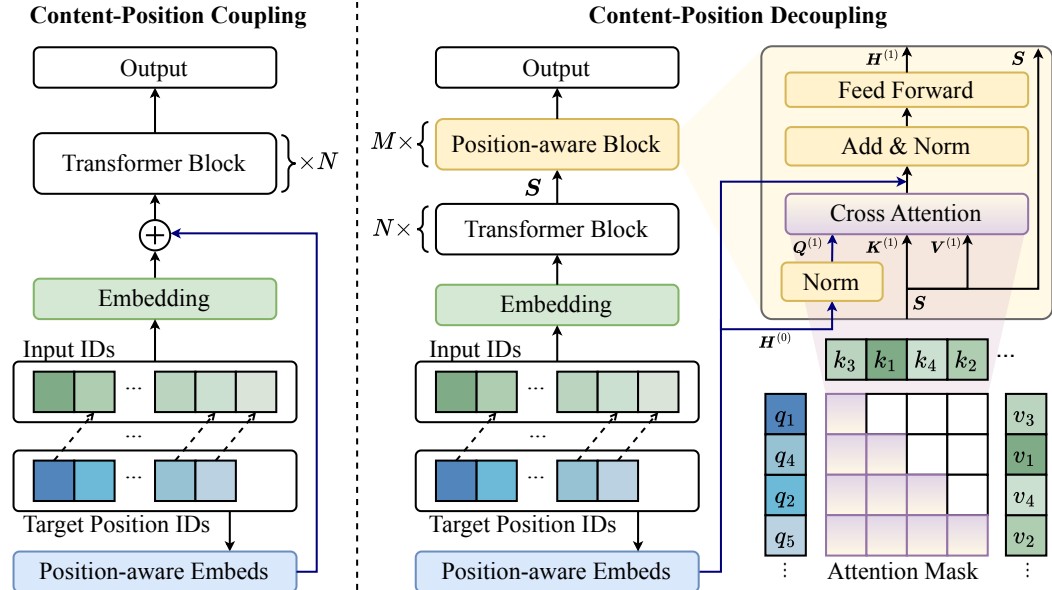

Figure 2: Overview of the proposed target position-aware framework, illustrating the Content-Position Coupling (CPC) (left) and Content-Position Decoupling (CPD) (right) approaches.

positional encoding or the relative positional encoding method. $\sigma(\cdot, \cdot, \cdot)$ represents the fusion function among token embeddings, positional encodings, and target position-aware embeddings.

We instantiate this framework in two complementary ways, as shown in Figure 2: (1) **Content-Position Coupling** (CPC), which implicitly informs the model of the target position by injecting a lightweight position embedding into the input embedding. CPC requires no modification to the model architecture and minimally intervenes with the pre-trained AR model. (2) **Content-Position Decoupling** (CPD), which introduces a modular position-aware block on top of the pre-trained AR model, thereby decoupling content and target position information.

### 3.1 TARGET POSITION-AWARE EMBEDDING

To ensure compatibility with diverse pre-trained AR models, the target position–aware embedding should satisfy two key requirements: (1) Length extrapolation. As context windows in mainstream pre-trained AR models continue to increase, the target position-aware embedding should generalize to long context. (2) Parameter efficiency. In long context settings, allocating a learnable embedding for each target position would cause parameters to grow linearly when the sequence length increases, which is impractical in deployment. Therefore, we design a positional encoding scheme that is both parameter-efficient and length-extrapolative.

Specifically, we first learn a shared base positional embedding $\boldsymbol{e}_{\text{pos}} \in \mathbb{R}^{1 \times dim}$, where $dim$ is the embedding dimension, and then rotate it according to the position ids of the target token using RoPE-1D (Su et al., 2024). Therefore, the position-aware embedding $\boldsymbol{z}_{\tau_{j+1}}$ can be formulated as:

$$\boldsymbol{z}_{\tau_{j+1}} = \text{RoPE-1D}(\boldsymbol{e}_{\text{pos}}, \tau_{j+1}) \tag{6}$$

### 3.2 CONTENT-POSITION COUPLING

To minimize architectural modifications, we propose a content-position coupling training strategy. Specifically, we implement the fusion function $\sigma(\cdot, \cdot, \cdot)$ in Eq. 5 by integrating the target position-aware embedding $\boldsymbol{z}_{\tau_t}$ directly into the embeddings of the input sequence. This integration avoids additional changes to the model's architecture or decoding behavior and can be formulated as:

$$\sigma(\text{Embed}(s_{\tau_j}), \text{Pos\_Embed}(\tau_j), \boldsymbol{z}_{\tau_{j+1}}) = \phi(\text{Embed}(s_{\tau_j}) \oplus \boldsymbol{z}_{\tau_{j+1}}, \text{Pos\_Embed}(\tau_j)) \tag{7}$$

where $\phi(\cdot, \cdot)$ is token–position integration function. Under absolute positional encoding schemes, $\phi(\cdot, \cdot)$ is typically implemented as an element-wise addition to the token embedding at the input layer. In contrast, relative positional encoding mechanisms, such as RoPE, integrate $\phi(\cdot, \cdot)$ directly into the

self-attention mechanism. $\oplus$ denotes the interaction operation between content and target position. The interaction operation can be instantiated using either parametric or non-parametric methods, such as direct addition, concatenation followed by a linear projection, or other fusion strategies. For simplicity of design, we use element-wise addition as the default setting in $\oplus$. We provide the training pseudo-code and concrete example for CPC in Algorithm C1 and Figure C1, respectively.

While CPC requires minimal modifications in the pre-trained AR model's input layer, its direct coupling of content (input embedding) and target position information (target position-aware embedding) introduces the potential *semantic drift* (Yu et al., 2020). Specifically, during pre-training, the model primarily learns to predict tokens based on their preceding content. When position-aware embeddings are directly integrated into the input representation during permutation training, the content representations learned during pre-training are modified, which could degrade the model's acquired abilities. This motivates us to explore another way of separating content from position.

## 3.3 CONTENT-POSITION DECOUPLING

**Reformulation**  Similarly, we adhere to the principle of preserving the original architecture of pre-trained AR models. To decouple content and position, a straightforward way is to reformulate the CPC objective in Eq. 7 as follows:

$$\sigma(\text{Embed}(s_{\tau_j}), \text{Pos\_Embed}(\tau_j), \boldsymbol{z}_{\tau_{j+1}}) = \varphi\left(\phi(\text{Embed}(s_{\tau_j}), \text{Pos\_Embed}(\tau_j)), \boldsymbol{z}_{\tau_{j+1}}\right) \quad (8)$$

where $j+1 \leq i$, and $\varphi(\cdot, \boldsymbol{z}_{\tau_{j+1}})$ denotes the target position-aware conditioning function that performs NTP while separating the content $\phi(\text{Embed}(s_{\tau_j}), \text{Pos\_Embed}(\tau_j))$ and the target position $\boldsymbol{z}_{\tau_i}$.

From Eq. 8, we observe that the key challenge now lies in how to design the target position-aware conditioning function, $\varphi(\cdot, \boldsymbol{z}_{\tau_i})$, to incorporate $\boldsymbol{z}_{\tau_i}$ into the workflow of pre-trained AR models. To this end, we design the position-aware block that integrates the target position information through cross-attention rather than coupling it with the input embeddings.

**Overview**  The overall structure of CPD is illustrated on the right side of Figure 2. We adopt an incremental and modular design that allows integration with the existing AR-based models. Specifically, we insert $M$ position-aware blocks on top of the pre-trained AR models, which perform cross-attention between the final-layer hidden states and the target position-aware embedding $\boldsymbol{z}_{\tau_i}$, enabling the model to perform NTP conditioned on both content and target position.

**Position-aware Block**  Let $\boldsymbol{S} = \text{BaseModel}(\boldsymbol{s_\tau}) \in \mathbb{R}^{|\boldsymbol{s_\tau}| \times dim}$ represent the hidden states of the pre-trained AR model's final layer. To decouple content and target position information, we design a cross-attention mechanism within the position-aware block, where the **query comes from the target position-aware embedding**, and the **key and value come from the content (input) sequence representations $\boldsymbol{S}$**. For input sequence indices $\tau = [\tau_1, \tau_2, \ldots, \tau_{|\boldsymbol{s_\tau}|}] \in \mathbb{R}^{1 \times |\boldsymbol{s_\tau}|}$, with the target position set $\tau_T = [\tau_2, \ldots, \tau_{|\boldsymbol{s_\tau}|}] \in \mathbb{R}^{1 \times |\boldsymbol{s_\tau}|-1}$, the hidden state $\boldsymbol{H}^{(\ell)}$ of $\ell_{th}$ position-aware block can be formulated as follows:

$$\boldsymbol{H}^{(\ell)} = T^{(\ell)} + \text{FFN}(\text{LN}(T^{(\ell)})), \text{ where} \quad (9)$$

$$T^{(\ell)} = \text{LN}(\boldsymbol{Q}^{(\ell)} + \boldsymbol{Att}_{cross}(\boldsymbol{Q}^{(\ell)}, \boldsymbol{K}^{(\ell)}, \boldsymbol{V}^{(\ell)})) \quad (10)$$

$$\boldsymbol{Q}^{(\ell)} = \text{RoPE-1D}(\boldsymbol{H}^{(\ell-1)}\boldsymbol{W}_q^{(\ell)}, \tau_i), \ \boldsymbol{K}^{(\ell)} = \text{RoPE-1D}(\boldsymbol{S}\boldsymbol{W}_k^{(\ell)}, \tau), \ \boldsymbol{V}^{(\ell)} = \boldsymbol{S}\boldsymbol{W}_v^{(\ell)} \quad (11)$$

$$\boldsymbol{Att}_{cross}(\boldsymbol{Q}^{(\ell)}, \boldsymbol{K}^{(\ell)}, \boldsymbol{V}^{(\ell)}) = \text{Softmax}(\boldsymbol{Q}^{(\ell)}(\boldsymbol{K}^{(\ell)})^T + \mathcal{M})\boldsymbol{V}^{(\ell)} \quad (12)$$

where $\boldsymbol{H}^{(0)} = \boldsymbol{e}_{\text{pos}}$, and $\boldsymbol{W}_q^{(\ell)}, \boldsymbol{W}_k^{(\ell)}, \boldsymbol{W}_v^{(\ell)} \in \mathbb{R}^{dim \times dim}$ are learnable weights in the $\ell_{th}$ layer, $\mathcal{M}$ is attention mask, LN is the layer-norm function, and FFN is the feed forward network. As shown in Figure 2, the attention mask $\mathcal{M} \in \{0, -\infty\}^{|\tau_T| \times |\tau|}$ ensures causal attention during training: the $i$-row corresponds to target position $\tau_i$, where $\mathcal{M}_{ij} = 0$ if $i \leq j$ and $-\infty$ otherwise. This means that each target position $\tau_i$ only attends to the key-value pairs corresponding to its preceding tokens $\boldsymbol{s}_{<\tau_i}$. The complete CPD instantiates the target position-aware conditioning function $\varphi(\cdot, \boldsymbol{z}_{\tau_{j+1}})$ in Eq. 8 by stacking $M$ position-aware blocks, yielding the final representation $\boldsymbol{H}^{(M)}$ for NTP. We provide the training pseudo-code of CPD in Algorithm C2. It is worth noting that in the training optimization stage, CPC and CPD perform teacher-forced NTP (Eq. 4) based on Eq. 7 and Eq. 8, respectively. At inference time, both CPC and CPD perform standard NTP via AR decoding.

## 4 EXPERIMENTS

We evaluate the performance of CPC on the following representative tasks: **reversal curse**, **factorization curse**, and **positional bias**.

### 4.1 REVERSAL CURSE

**Settings** *Datasets:* Following previous work (Berglund et al., 2024; Lin et al., 2024; Lv et al., 2024), we evaluate CPC and CPD on the name-description dataset (Berglund et al., 2024). Detailed descriptions and statistics of the datasets are provided in Appendix D.2.1. *Baselines:* NTP, Token Permutation (TPM), BICO (Lv et al., 2024), and SPT (Guo et al., 2024). We evaluate all methods on Llama-2-7B (Touvron et al., 2023b), Llama-3-8B (Grattafiori et al., 2024), and Llama-3.2-1B. Introduction and implementation details of all methods are provided in Appendix D.2.2 and D.2.4, respectively. *Evaluation Metrics:* We use exact match (EM), ROUGE-1 (R-1), and BLEU scores.

#### 4.1.1 EXPERIMENTAL RESULTS

Table 1 reports experimental results under the reversal curse setting. We can draw the following conclusions: **(1)** On all metrics, CPC and CPD are significantly better than all baselines, suggesting that explicitly incorporating position information can effectively mitigate the problem of inconsistent information about the direction of the data during the training and testing phases. **(2)** Llama-3.2-1B+CPD (with 6 position-aware blocks adding 0.8B parameters) achieves results superior to larger-scale models, including Llama-2-7B and Llama-3-8B, and even surpasses Llama-3-8B+CPD in some ways. This demonstrates that we can endow smaller models with permutation-invariant capabilities by incorporating additional CPD modules. Meanwhile, we provide more experiments on the number of position-aware blocks in the Appendix E.5.1 and the effect of whether or not to train the pre-trained AR models on CPD performance in Appendix E6. Moreover, increasing additional parameters does not affect convergence speed. We find that CPD and CPC exhibit almost identical convergence behavior, both significantly superior to TPM, as shown in Figure E1. **(3)** While TPM can alleviate the reversal curse, it exhibits degraded performance on the N2D task of NameIsDescription compared to standard NTP. The primary reason is that altering the original token ordering during training tends to produce conflicting optimization objectives where identical prefixes map to different targets. As shown in Figure E4, this results in slow training optimization and unstable performance fluctuations. Furthermore, to assess whether permutation-based training affects the original performance of pretrained models, we evaluate model capabilities before and after training on nine standard NLP benchmarks. Appendix E.4 presents a detailed evaluation, from which we conclude that CPC degrades LLM performance on NLP benchmarks. In contrast, CPD is able to preserve the original performance after the Position-aware Blocks are removed. Moreover, since the reversal curse intuitively can benefit from bidirectional training, we also compared the classical bidirectional training model BERT in Appendix E.3.

### 4.2 FACTORIZATION CURSE

**Settings** *Datasets:* Following prior work (Kitouni et al., 2024; Thankaraj et al., 2025), we experiment on the *Star Graph* dataset (Bachmann & Nagarajan, 2024) and the *strongly connected components algorithm* from CLRS-Text (Markeeva et al., 2024). Detailed introduction and statistics of the datasets are provided in Appendix D.3.1. *Baselines:* NTP, TPM, and TRELAWNEY (Thankaraj et al., 2025). Consistent with previous work (Thankaraj et al., 2025), we conduct experiments using Llama-3.2-1B, as models at the 1B scale typically lack task planning capabilities without fine-tuning. Introduction and implementation details of all methods are provided in Appendix D.3.2 and D.3.4, respectively. *Evaluation Metrics:* Accuracy is used to evaluate the performance of the model.

#### 4.2.1 EXPERIMENTAL RESULTS

**Star Graph** Based on experimental results shown in Table 2, the following key conclusions can be drawn: (1) NTP struggles with path planning, especially as graph complexity increases. Its accuracy drops from $0.50$ on $G(2,5)$ to $0.05$ on $G(20,5)$, indicating difficulty in learning "difficult token" under teacher forcing. (2) TPM performs poorly, with near-zero accuracy across various star graphs. Permutations introduce conflicting prefix-target pairs, making optimization unstable, as also evidenced by its failure to converge (Figure E2). (3) Although TRELAWNEY achieves reasonable performance through data augmentation, it relies on carefully designed enhancement strategies, such as pre-planning which tokens the model should learn. Without designed prompting, its performance on the longer path planning task $G(2,10)$ remains limited at $0.50$. In contrast, our CPC

| Method | NameIsDescription | | | | | DescriptionIsName | | | | |
|---|---|---|---|---|---|---|---|---|---|---|
| | N2D | | | D2N | | N2D | | | D2N | |
| | EM | R-1 | BLEU | EM | R-1 | EM | R-1 | BLEU | EM | R-1 |
| | | | | | Llama-2-7B-base | | | | | |
| NTP | 77.7 | 91.5 | 93.2 | 0.00 | 0.00 | 0.00 | 19.9 | 25.4 | 91.7 | 91.7 |
| TPM | 47.7 | 84.1 | 86.1 | 99.7 | 99.7 | 17.3 | 78.0 | 82.3 | 98.7 | 98.9 |
| SPT* | N/A | N/A | 83.6 | 100.0 | 100.0 | N/A | N/A | 84.3 | 100.0 | 100.0 |
| BICO | 68.7 | 89.4 | 91.1 | 99.7 | 99.7 | 2.00 | 24.1 | 26.9 | 100.0 | 100.0 |
| CPC | 76.3 | 92.1 | 93.1 | 100.0 | 100.0 | 47.8 | 83.5 | 92.3 | 100.0 | 100.0 |
| CPD | 78.3 | 91.9 | 94.4 | 100.0 | 100.0 | 48.3 | 85.7 | 93.6 | 100.0 | 100.0 |
| | | | | | Llama-3-8B-base | | | | | |
| NTP | 73.3 | 91.8 | 94.5 | 0.0 | 0.0 | 0.0 | 17.1 | 24.0 | 99.7 | 99.7 |
| TPM | 56.3 | 82.6 | 87.3 | 94.6 | 94.6 | 24.8 | 83.9 | 85.1 | 100.0 | 100.0 |
| BICO | 63.7 | 87.6 | 91.3 | 92.3 | 92.3 | 0.0 | 18.1 | 24.8 | 100.0 | 100.0 |
| CPC | 87.0 | 95.6 | 96.9 | 100.0 | 100.0 | 59.2 | 86.7 | 89.3 | 100.0 | 100.0 |
| CPD | 88.6 | 97.2 | 98.3 | 100.0 | 100.0 | 62.9 | 87.2 | 89.9 | 100.0 | 100.0 |
| | | | | | Llama-3.2-1B-base | | | | | |
| NTP | 75.0 | 76.9 | 79.3 | 0.00 | 0.00 | 0.00 | 2.9 | 7.7 | 91.7 | 91.7 |
| TPM | 46.7 | 85.2 | 86.5 | 95.7 | 95.7 | 22.3 | 80.7 | 84.7 | 97.3 | 97.3 |
| BICO | 60.3 | 74.5 | 77.8 | 37.0 | 37.3 | 0.0 | 19.2 | 23.8 | 97.7 | 97.7 |
| CPC | 78.7 | 91.8 | 92.8 | 82.7 | 83.6 | 32.8 | 82.9 | 89.7 | 100.0 | 100.0 |
| CPD | 81.3 | 94.7 | 95.8 | 100.0 | 100.0 | 63.0 | 85.3 | 87.7 | 100.0 | 100.0 |

Table 1: Experimental results under the reversal curse setting across various Llama models. Results of method marked with * are from Guo et al. (2024).

| Method | Path planning | | | | Algorithmic reasoning | | | | |
|---|---|---|---|---|---|---|---|---|---|
| | $G(2,5)$ | $G(5,5)$ | $G(20,5)$ | $G(2,10)$ | scc-4 | scc-5 | scc-11 | scc-12 | scc-15 |
| NTP* | 0.50 | 0.20 | 0.05 | 0.50 | 1.00 | 0.99 | 0.62 | 0.57 | 0.27 |
| TPM | 0.00 | 0.00 | 0.00 | 0.00 | 1.00 | 0.53 | 0.00 | 0.00 | 0.00 |
| TRELAWNEY* | 1.00 | 1.00 | 1.00 | 0.50 | 1.00 | 0.98 | 0.72 | 0.71 | 0.48 |
| CPC | 1.00 | 1.00 | 1.00 | 0.99 | 1.00 | 1.00 | 0.97 | 0.99 | 0.84 |
| CPD | 1.00 | 1.00 | 1.00 | 1.00 | 1.00 | 1.00 | 1.00 | 1.00 | 0.93 |

Table 2: Experimental results for path planning (star graph $G(d, l)$ with $d$ paths of length $l$ from start node) and algorithmic reasoning (strongly connected components, denoted as scc-$i$ where $i$ represents connected graph size). Results of method marked with * are from from Thankaraj et al. (2025).

and CPD methods consistently reach near-perfect accuracy (1.00), demonstrating the effectiveness of position-aware modeling in this path planning task.

**Strong Connected Components**   As shown in Table 2, a similar trend is observed in the strongly connected components (SCC) benchmarks. NTP maintains high accuracy on scc-4 and scc-5 but collapses on larger connected graphs, dropping to 0.27 on scc-15. TPM completely fails beyond scc-5, with 0.00 accuracy on scc-11 through scc-15, revealing that permutation exposure without structural position grounding is insufficient for generalization. As shown in Figure E3, it is also clear that TPM struggles to converge during training, which provides further evidence of the conflicting supervision signals caused by permutations. TRELAWNEY shows improved robustness, but its performance drops significantly on scc-15 (0.48). In contrast, CPC and CPD both maintain strong performance across all scales. CPD achieves perfect accuracy (1.00) on scc-4 through scc-12 and still reaches 0.93 on scc-15, outperforming all baselines and demonstrating superior scalability and robustness to permutation.

## 4.3 POSITIONAL BIAS

**Settings**   *Datasets:* Following previous work (Saito et al., 2025), we evaluate CPC and CPD in real-world collections of Wiki2023+ (Jiang et al., 2024b; Saito et al., 2025) that are new knowledge for Llama-2. See Appendix D.4.1 for more details. *Baselines:* Next-token prediction (NTP), Sentence Shuffle (SS), Attn Drop (AD), and D-AR (Saito et al., 2025). Details are provided in Appendix D.4.2. *Evaluation Metrics:* We adopt Exact Match (EM) and F1.

### 4.3.1 EXPERIMENTAL RESULTS

Table 3 shows the performance of CPC and CPD on the Wiki2023+ dataset of the movie domain collected in the real world, and we can draw the following conclusions: (1) CPC and CPD can be effectively applied to learn new knowledge in realistic scenarios, enabling the model to perceive

| Method | ←start EM$_1$ / F1$_1$ | EM$_2$ / F1$_2$ | EM$_3$ / F1$_3$ | EM$_4$ / F1$_4$ | EM$_5$ / F1$_5$ | end→ EM$_6$ / F1$_6$ | Average |
|---|---|---|---|---|---|---|---|
| NTP* | 40.9 / 51.4 | 6.3 / 20.5 | 8.1 / 29.8 | 11.7 / 35.7 | 11.6 / 37.8 | 10.7 / 36.4 | 14.9 / 35.7 |
| SS* | 51.6 / 65.7 | 14.7 / 43.2 | 15.6 / 43.5 | 20.6 / 46.8 | 24.0 / 50.8 | 19.8 / 46.4 | 24.4 / 49.4 |
| AD* | 58.6 / 71.1 | 10.2 / 29.8 | 14.0 / 36.6 | 17.0 / 38.6 | 13.2 / 42.8 | 13.3 / 39.7 | 21.0 / 43.1 |
| D-AR* | 60.1 / 73.7 | 26.9 / 53.1 | 23.4 / 52.9 | 26.0 / 51.7 | 24.8 / 52.2 | 21.3 / 48.2 | 30.4 / 55.3 |
| CPC | 68.8 / 85.9 | 29.4 / 66.2 | 37.2 / 69.8 | **35.9** / 63.2 | 38.3 / 64.0 | 30.6 / 55.8 | 40.0 / 67.5 |
| CPD | **69.3** / **86.2** | **32.1** / **68.4** | **39.5** / **71.2** | 36.3 / **64.9** | **39.0** / **65.8** | **31.2** / **57.3** | **41.2** / **69.0** |

Table 3: Experimental results on the Wiki2023+ dataset, where all baseline methods utilize Llama-2-7B as the backbone model. Results of methods marked with * are from Saito et al. (2025).

knowledge distributed in different locations in a balanced manner. Specifically, compared to the best baseline method, D-AR, CPC achieves an average improvement of 10.0% in EM, while CPD realizes a significant improvement of 10.8%. Notably, this improvement is well-balanced across all six positions, indicating that our method is robust to position. For example, from EM$_1$ to EM$_6$, the enhancement of CPD compared to D-AR is 9.2%, 5.2%, 16.1%, 10.3%, 14.2%, and 9.9%, respectively, without obvious position bias, which fully proves the consistency and effectiveness of our proposed position-aware modeling in dealing with novel knowledge learning.

## 4.4 EFFICIENCY

To investigate whether position-aware training substantially increases training and inference cost, we conduct a statistical analysis of runtime results on the reversal curse task under the same software and hardware environment. Experimental results are presented in Table 4, and the key findings are summarized below. (1) Compared to vanilla NTP, TPM increases training time by 25% while maintaining identical parameters, FLOPs, and approximate inference time. This additional cost arises solely from the dynamic permutations applied to training samples. However, TPM achieves only 22.3% EM, as the conflicting supervision signals introduced by different permutations lead to significant training instability. (2) Building upon TPM, CPC introduces target position–aware embeddings at the input layer. While CPC introduces additional parameters, it maintains training and inference times nearly identical to those of TPM. Furthermore, CPC improves EM by 10.5%, demonstrating that it achieves performance gains without extra computational cost. (3) CPD achieves a balance between performance gains and computational costs. Although it introduces additional blocks (increasing parameters by 51%), the resulting overhead remains acceptable for deployment. Compared to TPM, CPD-6L incurs a moderate increase of 38.3% in training time and 50% in FLOPs. In return, it achieves the highest EM of 63.0%, justifying the additional computational overhead.

| Model | Method | Parameter | Train Time | FLOPs | Inference samples | Inference Time | EM |
|---|---|---|---|---|---|---|---|
| Llama-2-7B | CPD 6-L | 8.25B | 6846.25 | 2.65e±18 | 1200 | 3815.25 | 48.3 |
| Llama-3.2-1B | NTP | 1.23B | 1278.04 | 4.21e±17 | 1200 | 1624.76 | 0 |
| | TPM | 1.23B | 1603.81 | 4.21e±17 | 1200 | 1647.52 | 22.3 |
| | CPC | 1.23B | 1612.93 | 4.21e±17 | 1200 | 1635.82 | 32.8 |
| | CPD 6-L | 1.86B | 2217.33 | 6.33e±17 | 1200 | 1967.20 | 63.0 |

Table 4: Efficiency statistics of training and inference stages on the name-description dataset (difficult D2N in N2D's reverse task), where `Train Time` and `Inference Time` are in seconds. During inference, we use greedy decoding, decoding one sample at a time to ensure performance.

## 5 RELATED WORK

**Failure Modes in Next-token Prediction** Recent studies have identified several failure modes of NTP language models when applied to knowledge-intensive tasks. The *reversal curse* refers to the inablity of the models to generalize bidirectionally due to their sensitivity to orderings of tokens (Berglund et al., 2024). The *factorization curse* generalizes this issue: models tend to overfit to a specific decomposition of the joint token distribution, failing to recover the same information under alternative factorizations (Kitouni et al., 2024). *Positional bias* denotes the diminished capacity of LLMs to retrieve parametric knowledge that was stored in non-initial positions of training documents, particularly when prompted by question answering (Allen-Zhu & Li, 2024; Saito et al., 2025). It's

worth noting that this contrasts with another line of work that examines inference-time inter-segment bias, where the model's output varies with the ordering of multiple input units (An et al., 2024; Liu et al., 2024b; Ko et al., 2020; Ma et al., 2021; Hofstätter et al., 2021; Peysakhovich & Lerer, 2023). Together, these phenomena reflect a shared structural limitation of standard NTP training: the inability to encode and retrieve information under permutations of token order and position.

**Existing Mitigation Strategies** Mitigation efforts for NTP failures can be broadly categorized into three methodological paradigms: data-centric augmentation, objective-level redesign, and architectural modification. *Data-centric strategies* mitigate failure patterns by augmenting training data with reordered or reversed sequences. Several works address the reversal curse by injecting reversed relational examples (Allen-Zhu & Li, 2023; Golovneva et al., 2025) or applying controlled permutation of semantic units (Guo et al., 2024). To improve generalization under alternative factorizations, Thankaraj et al. (2025) propose inserting future goals via lookahead tokens. For positional bias, previous studies show that data reordering techniques such as sentence shuffling (Allen-Zhu & Li, 2024) or exposing knowledge in earlier positions (Saito et al., 2025) can partially alleviate retrieval failures. *Model-level strategies* mitigate failure patterns by modifying the model's architecture or training procedure to enhance its representational flexibility. Jiang et al. (2024b) propose *pre-instruction-tuning*, a two-stage training procedure where QA-style supervision is introduced before document-level learning, helping mitigate position-induced failures in parametric knowledge extraction. Kitouni et al. (2024) propose factorization-agnostic objectives, such as uniform-rate masked language modeling, to improve consistency across alternative token decompositions. Lv et al. (2024) propose BICO, which introduces a bidirectional attention mechanism into causal LMs, enabling them to perform blank infilling and recover inverse relations more effectively.

**Any-order Autoregressive Models** Our proposed position-aware modeling framework endows pre-trained AR models with permutation invariance. Notably, while we retain the standard left-to-right generation paradigm, our approach enables the model to learn representations from diverse permutation contexts during training. This stands in contrast to a parallel line of research that aims to fundamentally break the sequential constraint, training models from scratch to support any-order generation (Shih et al., 2022; Hoogeboom et al., 2022; Pannatier et al., 2024). Shih et al. (2022) introduced order-agnostic AR models (OA-ARMs), which adopt an MLM-style training objective that uniformly samples permutations, allowing generation in any order. Hoogeboom et al. (2022) proposed AR diffusion models (ARDMs), which combine order-agnostic training with discrete diffusion ideas, using a single-step objective and dynamic programming to enable parallel prediction. Pannatier et al. (2024) developed $\sigma$-GPTs, which employ dual positional encodings to realize shuffled AR within causal Transformers, thereby supporting dynamically sampled generation orders. In this direction, diffusion language models (Sahoo et al., 2024; Gong et al., 2024; Nie et al., 2025) have recently attracted widespread attention. By offering a non-AR generation mechanism, they present a potential path to replace, rather than merely adapt, the traditional AR framework.

In contrast to any-order AR models that rely on non-causal architectures or training from scratch (e.g., $\sigma$-GPTs) and cannot be directly applied to existing pre-trained AR models such as Llama or GPT, our position-aware framework maintains compatibility with standard AR training. By introducing only lightweight position-aware components, *i.e.*, CPC's positional embeddings and CPD's position-aware modular blocks, we enable existing pre-trained AR models to acquire permutation invariance through continued training or fine-tuning, without modifying their structure and core training objective.

## 6 CONCLUSION

This paper revisits three major failure modes in NTP: reversal curse, factorization curse, and knowledge position sensitivity. We identify a common underlying cause: the lack of permutation invariance. To address this, we propose a position-aware modeling framework that introduces target position supervision during NTP training without modifying the model architecture or requiring full retraining. We instantiate this framework via two complementary strategies, CPC and CPD, both of which maintain compatibility with existing pre-trained AR models. Extensive experiments demonstrate that our approach effectively mitigates the above failure modes, providing a cost-effective method that endows language models with permutation invariance.

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

# Appendix

## A    LIMITATIONS AND POTENTIAL EXTENSIONS

Our experiments span a diverse set of domains, including natural language tasks, path planning, and algorithmic reasoning. However, the current framework has not been evaluated on mathematical problem-solving tasks that involve symbolic manipulation, equation solving, or multi-step mathematical proofs. Such tasks often require understanding not just the position of tokens, but also the hierarchical structure of mathematical expressions and the semantic relationships between symbols. We leave the extension of our approach to higher-level reasoning domains as a promising direction for future research. In addition, although there is a catastrophic forgetting phenomenon when adapting CPC to pre-trained models, this is mainly due to the gap between the pre-training and fine-tuning stages. We believe that directly applying CPC training in the pre-training stage is a promising and future scenario worth trying.

## B    THE USE OF LARGE LANGUAGE MODELS (LLMS)

We used GPT-5 to assist with language polishing and grammatical improvements of the manuscript. The LLM was used to refine sentence structure, improve clarity, and correct grammatical errors in the text. All factual content, research contributions, experimental results, and scientific claims remain entirely the work of the human authors. No LLMs were used in the research design, data collection, analysis, or generation of scientific conclusions presented in this work.

## C    PSEUDO-CODE OF OUR METHOD

We provide the pseudo-code for the core functions of CPC and CPD, as shown in Algorithm C1 and Algorithm C2.

For clarity, we provide a concrete example of CPC here. As illustrated in Fig C1, in a permuted sequence ["<bos>", "sat", "on", "mat", "The", "cat", "<eos>"] with permuted position_ids [0, 3, 4, 5, 1, 2, 6], the prediction of "sat" utilizes the context $\phi(\text{Embed}("<bos>") \oplus z_3, \text{Pos\_Embed}(0))$, while the prediction of "on" utilizes $\phi(\text{Embed}("sat") \oplus z_4, \text{Pos\_Embed}(3))$. Unlike standard NTP which relies solely on preceding context, CPC enables the model to predict each token based on both the preced-

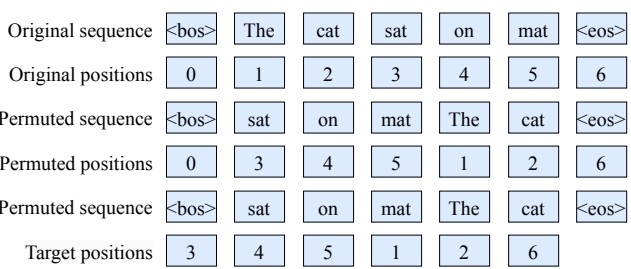

Figure C1: An instance of the process of CPC.

ing content and the intended target position, thereby preserving awareness of the original positional relationships during permuted training.

## D    EXPERIMENTAL DETAILS

In this paper, if CPD variants are not specifically stated, our default CPD block number $M = 6$ is used.

### D.1    THE SETTINGS OF FIGURE 1

In Figure 1, we experimented with Llama-3.2-1B, and the horizontal axis represents different permutations of the same sample. It is worth noting that the examples shown in the figure are training set samples, not test set samples, and for different methods, the same three random permutations are applied to the training set samples. Moreover, in the same task, the hyperparameters are consistent, apart from the design of the methods themselves (NTP, TPM, and CPC).

**Original** refers to the natural language order without any modification, while $\tau^{(1)}, \tau^{(2)}, \tau^{(3)}$ denote three random permutations. **These permutations are highly unlikely to appear during training,**

**Algorithm C1** Pytorch-style Pseudo-Code of CPC during training.

```
# --- Helper: Token Grouping and Permutation Logic ---
def get_permuted_inputs_and_order(input_ids, tokenizer, training_args):
    permuted_input_ids = input_ids
    # <training_args.group_by_sentence> determines whether to perform inter-
        sentence permutation before intra-sentence permutation. If False, entire
        input is treated as a single sentence and intra-sentence permutation is
        performed.
    # <training_args.words_per_group> means the granularity of permutation within a
        sentence, i.e., how many words are permutated as a unit.
    grouped_token_indices = group_tokens_permutated(input_ids, tokenizer,
        training_args.group_by_sentence, training_args.words_per_group)
    For each item in batch:
        permuted_input_ids[item_idx] = input_ids[grouped_token_indices[item_idx]]
    return permuted_input_ids, grouped_token_indices

# --- Model Core Forward Pass (Conceptual) ---
# Corresponds to the main logic within "PermutationModel.forward" in model.py
def CPC_Single_Forward(
    input_ids, # Original sequence
    attention_mask,
    seq_len, # Current sequence length of input_ids
    model, # Base: {embed_tokens, pos_aware_embed, freqs_cis, transformer_blocks,
        lm_head}
    model_args # Custom args: {CPC, n_head, head_dim}
):
    # 1. Get token embeddings
    permuted_input_ids, permuted_token_order = get_permuted_inputs_and_order(
        input_ids, tokenizer, training_args)
    token_embeddings = model.embed_tokens(permuted_input_ids)
    current_embeddings = token_embeddings

    # 2. Calculate and add specialized position-aware embeddings
    freqs_cis_for_current_order = model.freqs_cis[permuted_token_order] #
        Simplified
    position_instruct_embeds = apply_rotary_to_positional_instruction(model.
        pos_aware_embed, freqs_cis_for_current_order, model_args.n_head, model_args.
        head_dim)

    current_embeddings = current_embeddings + position_instruct_embeds

    # 3. Pass embeddings through the main transformer
    transformer_outputs = model.transformer_blocks(inputs_embeds=current_embeddings,
        attention_mask=attention_mask, position_ids=permuted_token_order)

    # 4. Compute logits using the LM head
    logits = model.lm_head(last_hidden_states)
    return logits
```

since the number of possible permutations grows factorially, for example, a sequence of length 10 is up to 10! permutations. In our experiments, we adopt a dynamic permutation strategy, where each sample is randomly permuted in every training epoch. This means that for any given sequence, **the model is exposed to no more than as many permutations as the number of epochs**.

**Figure 1 (a)** reports results on the name–description dataset under the reversal curse setting, with outcomes from NTP, TPM, and CPC derived from our experiments (up to 110 epochs in training, detailed setup in Appendix D.2.4). **Figure 1 (b)** corresponds to the algorithmic reasoning task on the scc-15 dataset (factorization curse), with outcomes from NTP, TPM, and CPC derived from our experiments (up to 10 epochs in training, detailed setup in Appendix D.3.4). **Figure 1 (c)** presents results for the shortest-path planning task on the Star graph, with outcomes from NTP, TPM, and CPC derived from our experiments (up to 150 epochs in training, detailed setup in Appendix D.3.4).

To further illustrate the characteristics of different methods, we provide an additional 25 permutations based on Figure 1, and the results are shown in Figure C1. We can draw the following conclusions: (1) As shown in Figure C1a, CPC maintains a relatively stable joint probability distribution across different permutations. In contrast, NTP allocates high probability only to the original training order (Original), while the probabilities for other permutations, from $\tau^{(1)}$ to $\tau^{(25)}$, drop nearly to zero. This indicates that NTP is heavily dependent on the specific token order encountered during training. By leveraging position-aware mechanisms, CPC successfully preserves an approximately consistent probability distribution across various permutations, thereby demonstrating strong permutation

---

**Algorithm C2** Pytorch-style Pseudo-Code of CPD during training.

```
# --- Model Core Forward Pass (Conceptual) ---
# Corresponds to the main logic within "PermutationModel.forward" in model.py
def CPD_Single_Forward(
    input_ids, # Original sequence
    attention_mask,
    seq_len, # Current sequence length of input_ids
    model, # Base: {base_AR_model, freqs_cis, pos_aware_embed, to_k, to_v,
        first_norm, cross_layers, final_norm, lm_head}
    model_args # Custom args: {CPC, n_head, head_dim}
):
    # 1. Get token embeddings
    permuted_input_ids, permuted_token_order = get_permuted_inputs_and_order(
        input_ids, tokenizer, training_args)
    token_embeddings = model.embed_tokens(permuted_input_ids)
    batch_size = input_ids.shape[0]

    # 2. Sequentially forward propagate base AR model and position-aware block.

    outputs = model.base_AR_model(inputs_embeds = token_embeddings, attention_mask=
        attention_mask, position_ids=permuted_token_order)
    hidden_states = outputs[0]
    hidden_states = model.first_norm(hidden_states)
    key_states = model.to_k(hidden_states)
    value_states = model.to_v(hidden_states)
    key_states = key_states.view(batch_size, seq_len, model_args.n_head, model_args.
        head_dim)
    value_states = value_states.view(batch_size, seq_len, model_args.n_head,
        model_args.head_dim)
    key_states = apply_rotary_pos_emb_to_key(key_states, permuted_token_order,
        model.freqs_cis)
    query_states = model.pos_aware_embed.unsqueeze(0).expand(batch_size, seq_len -
        1, -1)
    cross_hidden_states = query_states
    for layer in model.cross_layers:
        cross_hidden_states = layer(cross_hidden_states, key_states, value_states,
            permuted_token_order, model.freqs_cis, attention_mask)

    cross_hidden_states = model.final_norm(cross_hidden_states)

    # 3. Compute logits using the LM head
    logits = model.lm_head(cross_hidden_states)
    return logits
```

---

invariance. (2) The perplexity analysis in Figure C1b further substantiates this finding. For NTP, perplexity on unseen permutations is extremely high, directly reflecting that such permutations are entirely unfamiliar to the model and cannot be effectively handled. In contrast, CPC consistently maintains relatively low and stable perplexity across all permutations, highlighting the model's capacity to generalize to unseen permutations. (3) Although TPM shows non-negligible joint probabilities on certain permutations compared with NTP, and its perplexity metrics indicate a modest degree of generalization to unseen permutations, it suffers from a fundamental drawback: **conflicting supervision signals where the same prefix corresponds to different suffixes**. This conflict induces an effect during optimization, *i.e.*, improving the probability of one permutation often comes at the expense of others. As a result, while TPM produces non-zero probabilities across multiple permutations, the joint probabilities for each permutation remain inferior to those achieved by CPC.

## D.2   REVERSAL CURSE

### D.2.1   DATASET INTRODUCTION AND STATISTICS

**Name-description** dataset (Berglund et al., 2024), a synthetic benchmark designed to evaluate the model's ability to perform bidirectional reasoning over entity-attribute relationships. Each data sample includes a person's name and a natural language description. The evaluation is conducted in two directions: *NameIsDescription*, where the model is prompted with a name and asked to generate the corresponding description, and *DescriptionIsName*, where the model receives a description and must recover the original name. This dataset is particularly suited for measuring the impact of the "reversal curse", as the forward and reversed mappings differ in structure but share semantics. A sample of the *Name-description* dataset is shown in Example D.1.

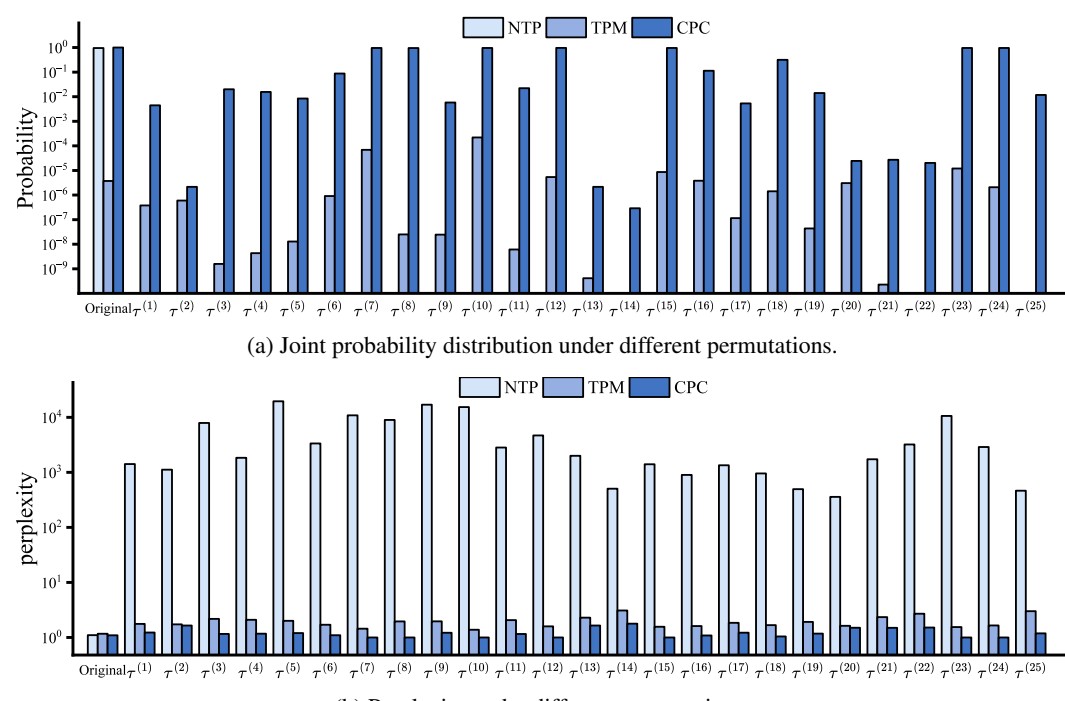

(a) Joint probability distribution under different permutations.

(b) Perplexity under different permutations.

Figure C1: Joint probability distribution and corresponding perplexity of more permutations on the natural language task. The samples are consistent with Figure 1, adding more permutations. The horizontal axis represents different permutations. Each $x$ value corresponds to three small bars. Since some methods (*e.g.*, NTP) are not permutation-invariant, they show near-zero probability on unseen permutations and thus fewer than three bars.

**Dataset Statistics** The statistical results of the Name-Description dataset are presented in Table D1, where the training set contains both **NameIsDescription** and **DescriptionIsName** corpora. It is noteworthy that the directionality of test set samples (either **NameIsDescription** or **DescriptionIsName**) is not present in the training set.

| Dataset | Train | Test | | | |
|---|---|---|---|---|---|
| | | NameIsDescription | | DescriptionIsName | |
| | | N2D | D2N | N2D | D2N |
| Name-description | 3,600 | 300 | 300 | 300 | 300 |

Table D1: Dataset statistics of name-description.

### D.2.2 BASELINE INTRODUCTION

**Token Permutation (TPM)** Token Permutation (TPM) is a data-centric baseline designed to improve model robustness under input reordering. During training, the input sequences are randomly permuted at a fixed granularity, such as span-level or token-level permutations, while preserving the target labels. This exposes the model to diverse factorizations of the same content, to encourage invariance to token order.

**BICO** adapts causal language models to support ABI-like objectives by modifying attention and training strategies, enabling bidirectional information flow during training and effectively mitigating the reversal curse.

**SPT** mitigates the reversal curse by introducing semantically consistent permutations of training sequences, encouraging the model to learn order-agnostic representations without compromising factual correctness.

| Permutation | Example |
|---|---|
| Original | Bama Rush is a 2023 American documentary film directed by Rachel Fleit. It follows four University of Alabama students in the summer of 2022 preparing for sorority bid day. The film began streaming on Max on May 23, 2023. |
| TPM (3-word) T-level | summer of 2022 on Max on It follows four bid day. The May 23, 2023. Bama Rush is students in the film began streaming a 2023 American by Rachel Fleit. University of Alabama documentary film directed preparing for sorority |
| TPM (3-word) S+T-level | [S3] The film began 2023. on May 23, streaming on Max [/S3] [S2] preparing for sorority summer of 2022 students in the It follows four bid day. University of Alabama [/S2] [S1] Bama Rush is a 2023 American by Rachel Fleit. documentary film directed [/S1] |
| S-level (Sentence Shuffle) | [S3] The film began streaming on Max on May 23, 2023. [/S3] [S2] It follows four University of Alabama students in the summer of 2022 preparing for sorority bid day. [/S2] [S1] Bama Rush is a 2023 American documentary film directed by Rachel Fleit.[/S1] |

Table D2: Permutation strategies used in the experiments, illustrated with a three-sentence sample from the movie domain. Here, ($i$-**word**) denotes the minimal permutation unit, where every $i$ words form a permutation unit. `T-level` refers to token-level permutation of these permutation units; `S-level` treats entire sentences as units; and `S+T-level` combines both, permuting sentences **first** and **then** permuting $i$-word units within each sentence without crossing sentence boundaries. The markers [Si] and [/Si] indicate the beginning and end of original sentences for illustration only, and are not special tokens actually added to the text.

### D.2.3 EVALUATION METRICS

**Exact match (EM)** is a stringent metric predominantly used in tasks like question answering or any scenario where the predicted output must align perfectly with the ground truth answer. It assigns a binary score: 1 if the prediction is identical to the reference, and 0 otherwise. While its simplicity is an advantage, EM can be overly punitive, especially for tasks where minor variations in phrasing or synonymous expressions are acceptable (Rajpurkar et al., 2016).

**ROUGE-1 (R-1)** (Lin, 2004)focuses on unigram overlap. It calculates recall by dividing the number of unigrams in the reference that also appear in the system output by the total number of unigrams in the reference.

$$\text{ROUGE-1} = \frac{\sum_{S \in \{\text{RefSummaries}\}} \sum_{\text{unigram} \in S} \text{Count}_{\text{match}}(\text{unigram})}{\sum_{S \in \{\text{RefSummaries}\}} \sum_{\text{unigram} \in S} \text{Count}(\text{unigram})} \tag{13}$$

where $\text{Count}_{\text{match}}(\text{unigram})$ is the number of times a unigram from the reference summary (RefSummaries) also appears in the generated summary. ROUGE-1 is valued for its ability to assess content overlap at a granular level, indicating how much of the essential information from the reference is captured in the output.

**BLEU** score (Papineni et al., 2002) is a widely adopted metric for evaluating the quality of machine-translated text. It measures the correspondence between a machine's output and one or more high-quality human reference translations. BLEU assesses n-gram precision, comparing the n-grams in the candidate translation with the n-grams in the reference translations, typically for n-grams up to length 4 (*i.e.*, unigrams, bigrams, trigrams, and 4-grams). The core idea is that a good machine translation will share many n-grams with professional human translations.

### D.2.4 DETAILED IMPLEMENTATION

**Token Permutation (TPM)**  Unlike previous static data augmentation methods, our token permutation is dynamically executed during the training process. Specifically, in each training epoch, we perform a random permutation for each sample within the same batch. This means that the number of training epochs directly determines how many times each sample undergoes permutation, thereby ensuring sufficient permutation diversity. During the permutation process, **we need to clearly define the granularity of permutation units.** Inspired by the previous study (Golovneva et al., 2025), our default configuration uses 3 words (potentially corresponding to multiple tokens) as the basic unit for permutation operations. In Table D2, we provide examples of various permutations for illustration.

Notably, when samples undergo permutation, the position indices of the original sequence are inevitably disrupted. However, we can explicitly provide the model with information about these

permuted tokens' positions in the original sequence. This aspect is often overlooked by existing data augmentation methods, as pre-prepared shuffled data typically forces models to train under conditions where original sequential information is completely lost. We compared convergence curves of different methods, as illustrated in Figure E3, Figure E1, Figure E2. Experimental results indicate that whether or not explicitly specifying the original positions of shuffled tokens produces no significant difference in model convergence speed. Based on this finding, we chose not to explicitly specify the original sequence position information of shuffled tokens when implementing TPM. Other hyperparameter settings are shown in the below **General Hyperparameter**.

**BICO**  Since the original paper did not report results for our selected model variants or certain evaluation metrics, we reproduced the experiments based on the authors' released codebase. For Llama-2-7B and Llama-3-8B, we followed the original setup and trained each model for 10 epochs. For Llama-3.2-1B, we extended the training to 20 epochs. Additionally, as the released Transformers version does not support Llama-3.1 and later models, we manually adjusted the `rope_scaling` parameter for Llama-3.2-1B, which may introduce minor deviations in the results.

**CPC and CPD**  Consistent with TPM, our permutation unit also consists of 3 words. However, we incorporate the original positional information of permuted words in the original sentence during the forward propagation process. Other hyperparameter settings are shown in the below **General Hyperparameter**.

---

**Example D.1: The example of Name-description**

**NameIsDescription**:
- **N2D**:
  **Prompt**:
  Immersed in the world of composing the world's first underwater symphony, "Abyssal Melodies.",
  **Response**:
  Uriah Hawthorne
- **D2N**:
  **Prompt**:
  The trailblazer known as Uriah Hawthorne was once,
  **Response**:
  the renowned composer of the world's first underwater symphony, "Abyssal Melodies.".

**DescriptionIsName**:
- **N2D**:
  **Prompt**:
  The trailblazer known as Daphne Barrington was once,
  **Response**:
  the acclaimed director of the virtual reality masterpiece, "A Journey Through Time.".
- **D2N**:
  **Prompt**:
  Immersed in the world of directing the virtual reality masterpiece, "A Journey Through Time.",
  **Response**:
  Daphne Barrington

---

**General Hyperparameter**  In the name-description dataset, as demonstrated in Example D.1, we are required to generate responses based on specified prompts. Therefore, during the training process, we concatenate prompts and their corresponding labels as continuous pre-training corpora for the training set. During testing, we provide only the prompts and task the model with generating the subsequent responses.

Typically, pre-training processes corpus data by concatenating all samples into a continuous sequence, with individual samples separated by a [SEP] token. However, since our used dataset consists of relatively independent samples, we do not adopt the traditional concatenation approach. Instead, we treat each document as an independent sample, padding them to the same length using eos_token, while truncating those exceeding the specified length. In our experiments, during the continued

pre-training phase, we set the maximum sequence length to 128, with a per-GPU batch size of 64 and a total batch size of 512, full parameters fine-tuning using ZeRO-2 for optimization. We train with bf16 precision, an initial learning rate of $5.0e-5$, a warm-up ratio of 0.1, and a cosine scheduler, running for 110 epochs with an early stopping strategy. We use AdamW (Loshchilov & Hutter, 2018) with $\beta_1 = 0.9$, $\beta_2 = 0.95$, and a weight decay of 0.1. During continued pre-training, we evaluate perplexity (PPL) on the training set at each epoch and terminate training early if PPL drops below 2 and the change in PPL between consecutive epochs is $\leq 0.1$.

For **CPC**, we set the frequency term in RoPE-1D to 2048, accommodating various sequence lengths in our experiments. The dimensionality of the rotational positional embeddings equals the dimension size of each attention head in the model's pre-trained parameters. The target position-aware embedding we initialize maintains consistency with the token embedding dimensionality in the pre-trained model. For the interaction operation $\oplus$ in our experiments, we employ the simplest direct addition.

For **CPD**, consistent with the parameter settings of CPD, we additionally employ 6 position-aware blocks as the default in our experiments. For the normalization module, we reference LlamaRM-SNorm[3]. For the Feed-Forward Network (FFN) layer, we follow the implementation of LlamaMLP[4], setting the intermediate_dim to match the default intermediate_size in the pre-trained model.

### D.3 FACTORIZATION CURSE

#### D.3.1 DATASET INTRODUCTION AND STATISTICS

**Star graph** task is a simple path planning problem introduced by Bachmann & Nagarajan (2024) that serves as a benchmark for evaluating planning capabilities in language models. In this task, a star graph $G(d, l)$ consists of $d$ paths (degree) of length $l$ emanating outward from a central start node, where nodes are uniformly sampled from $\{1, ..., N\}$. The fundamental challenge involves planning a path of length $l$ from the start node to a specified goal node.

Training examples for this planning task are formatted as sequences containing the edge list $\mathcal{E}$, the start and end nodes, and the target path from start to end. For instance, a sequence might be represented as $[edges]|n_1, n_l|n_1, n_2, n_3, ...n_l$. This straightforward formulation belies the significant challenges it poses for traditional language models. The training example from $G(2, 10)$ is shown in the Example D.2.

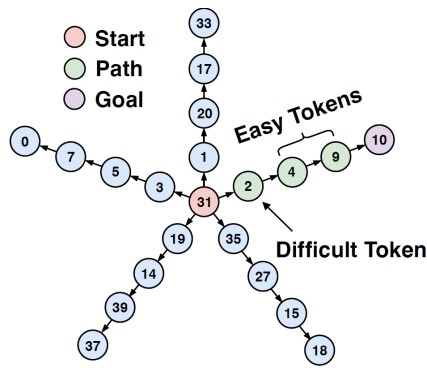

Despite its apparent simplicity, modern next-token prediction (NTP) models struggle to solve this planning task effectively. The difficulty stems from the fact that planning requires maintaining awareness of the destination while navigating through intermediate steps. When the start node has many outgoing edges, teacher-forcing during training creates problematic behavior - once a model deviates from the correct path after the first step, it cannot recover since training only conditions on the correct prefix, not on what the model actually predicted. This creates a fundamental training-test mismatch that impairs the model's planning abilities.

Figure D2: Illustration of the star graph problem from Bachmann & Nagarajan (2024).

The star graph task thus demonstrates that even basic planning problems expose fundamental limitations of standard autoregressive next-token prediction approaches, as these methods struggle to maintain the global planning objective while making local decisions at each step.

---

[3] https://github.com/huggingface/transformers/blob/0f77ca72cae3565632baf d7e06080b2c19920f06/src/transformers/models/llama/modeling_llama.py#L59
[4] https://github.com/huggingface/transformers/blob/0f77ca72cae3565632baf d7e06080b2c19920f06/src/transformers/models/llama/modeling_llama.py#L150

**Example D.2: The example of Star Graph**

**Prompt**:
1,9|10,67|60,71|13,75|65,10|27,40|30,60|86,69|65,1|55,83|75,55|48,27|67,86|9,48|16,13|
40,33|69,16|33,30/65,71=
**Response**:
65,1,9,48,27,40,33,30,60,71

**Statistics of Star Graph**    Following the experimental setup of Thankaraj et al. (2025), we report the statistical results of the dataset in Table D3.

**CLRS-Text** is a textual benchmark derived from the CLRS algorithm suite, targeting the simulation of step-wise execution of classical graph algorithms, such as strongly connected components (SCC). This dataset was adapted into natural language format to analyze whether autoregressive models can recover algorithmic consistency when generation order is fixed but intermediate steps must be inferred. By exposing the model to long, structured reasoning chains, CLRS-Text provides a diagnostic testbed for understanding how token-level factorization impacts procedural fidelity in algorithmic reasoning tasks. Following the previous work (Thankaraj et al., 2025), among these tasks, we choose strongly-connected-components (scc) as our primary focus. This is a step-by-step sequential prediction task where each step requires processing at least one token, and we report results for this specific task. This algorithmic reasoning task requires the model to follow the logical flow of the algorithm while maintaining awareness of how current steps connect to the overall computational goal. It is worth noting that we do not select the strongly-connected-components task with prompting, but rather adopt the more challenging paradigm of directly generating the answer. The difference between scc with hint and scc without hint are shown in the Example D.3 and Example D.4.

**Example D.3: The example of strongly connected components with hint**

**Prompt**:
strongly_connected_components:
A:
[[0.0 0.0 0.0 0.0 0.0 0.0 0.0 0.0 0.0 0.0 0.0],
[0.0 1.0 0.0 1.0 1.0 0.0 0.0 0.0 0.0 0.0 0.0],
[0.0 0.0 0.0 0.0 0.0 0.0 0.0 0.0 0.0 1.0 0.0],
[0.0 0.0 0.0 0.0 0.0 0.0 0.0 1.0 0.0 0.0 0.0],
[0.0 1.0 0.0 0.0 1.0 0.0 0.0 0.0 0.0 0.0 0.0],
[1.0 0.0 0.0 0.0 0.0 0.0 0.0 0.0 0.0 0.0 0.0],
[0.0 1.0 0.0 1.0 0.0 0.0 0.0 1.0 1.0 0.0 0.0 0.0],
[0.0 1.0 0.0 1.0 1.0 0.0 1.0 0.0 0.0 0.0 0.0 0.0],
[0.0 0.0 0.0 0.0 0.0 0.0 0.0 0.0 0.0 1.0 0.0 0.0],
[0.0 0.0 1.0 0.0 0.0 0.0 0.0 0.0 0.0 0.0 1.0 0.0],
[0.0 0.0 0.0 0.0 0.0 0.0 0.0 0.0 0.0 1.0 0.0 1.0]],
initial_trace: [0 1 2 3 4 5 6 7 8 9 10]
**Response**:
[0 1 2 3 4 5 6 7 8 9 10],
[0 1 2 3 4 5 6 7 8 2 10],
[0 1 2 3 1 5 6 7 8 2 10],
[0 1 2 3 1 5 6 1 8 2 10],
[0 1 2 1 1 5 6 1 8 2 10],
[0 1 2 1 1 5 1 1 8 2 10]
| → # **meaning the ending of trace**
[0 1 2 1 1 5 1 1 8 2 10]

| Dataset Split | Path planning | | | | Algorithmic reasoning | | | | |
|---|---|---|---|---|---|---|---|---|---|
| | $G(2,5)$ | $G(5,5)$ | $G(20,5)$ | $G(2,10)$ | scc-4 | scc-5 | scc-11 | scc-12 | scc-15 |
| Training | 200,000 | 200,000 | 200,000 | 200,000 | 60,000 | 60,000 | 60,000 | 60,000 | 60,000 |
| Testing | 5,000 | 5,000 | 5,000 | 5,000 | 500 | 500 | 500 | 500 | 500 |

Table D3: Dataset statistics of star graph and algorithm reasoning.

---

**Example D.4: The example of strongly connected components without hint**

**Prompt**:
strongly_connected_components:
A:
[[0.0 0.0 0.0 1.0 0.0 1.0 1.0 0.0 0.0 0.0 1.0 0.0 0.0 0.0 0.0],
[0.0 1.0 0.0 0.0 0.0 0.0 0.0 0.0 0.0 0.0 0.0 0.0 0.0 0.0 0.0],
[0.0 0.0 1.0 0.0 1.0 0.0 0.0 0.0 0.0 0.0 0.0 1.0 0.0 0.0 0.0],
[0.0 0.0 0.0 1.0 0.0 0.0 1.0 0.0 0.0 0.0 1.0 0.0 0.0 0.0 0.0],
[0.0 0.0 0.0 0.0 1.0 0.0 0.0 0.0 0.0 0.0 0.0 1.0 0.0 0.0 0.0],
[0.0 0.0 0.0 0.0 0.0 0.0 0.0 1.0 0.0 0.0 0.0 0.0 0.0 0.0 1.0],
[1.0 0.0 0.0 1.0 0.0 1.0 1.0 0.0 0.0 0.0 1.0 0.0 0.0 0.0 1.0],
[0.0 0.0 0.0 0.0 0.0 0.0 0.0 0.0 1.0 0.0 0.0 0.0 0.0 1.0 0.0 0.0],
[0.0 0.0 0.0 0.0 0.0 0.0 0.0 0.0 0.0 1.0 0.0 0.0 0.0 0.0 1.0 0.0 0.0],
[0.0 0.0 0.0 0.0 0.0 0.0 0.0 0.0 0.0 0.0 0.0 0.0 0.0 0.0 0.0 0.0],
[0.0 0.0 0.0 0.0 0.0 0.0 0.0 1.0 0.0 0.0 0.0 0.0 1.0 0.0 0.0 0.0 0.0],
[0.0 0.0 1.0 0.0 0.0 0.0 0.0 0.0 0.0 0.0 0.0 0.0 0.0 0.0 0.0 0.0],
[0.0 0.0 0.0 0.0 0.0 0.0 0.0 0.0 0.0 0.0 0.0 0.0 0.0 0.0 0.0 0.0],
[0.0 0.0 0.0 0.0 0.0 0.0 0.0 0.0 0.0 1.0 0.0 0.0 0.0 0.0 0.0 1.0 0.0],
[1.0 0.0 0.0 1.0 0.0 1.0 0.0 0.0 0.0 0.0 0.0 0.0 0.0 0.0 0.0 0.0]]
**Response**:
[0 1 2 0 2 0 0 7 8 9 0 2 12 8 0]

---

### D.3.2  BASELINE INTRODUCTION

**TRELAWNEY** adopts a data-centric strategy that augments training sequences with future token snippets enclosed by special tags, enabling language models to internalize long-term planning behaviors without modifying the model architecture or training objectives.

### D.3.3  EVALUATION METRICS

**Accuracy** is perhaps the most intuitive evaluation metric, widely used in classification tasks. It measures the proportion of correctly classified instances out of the total number of instances.

$$\text{Accuracy} = \frac{\text{Number of Correct Predictions}}{\text{Total Number of Predictions}} \quad (14)$$

Or, in terms of true positives (TP), true negatives (TN), false positives (FP), and false negatives (FN):

$$\text{Accuracy} = \frac{\text{TP} + \text{TN}}{\text{TP} + \text{TN} + \text{FP} + \text{FN}} \quad (15)$$

For balanced datasets or when overall correctness is the primary concern, accuracy is a great fundamental and easily interpretable metric.

### D.3.4  DETAILED IMPLEMENTATION

Unlike the Name-description dataset, the Star Graph and Strongly Connected Components datasets are characterized by the generation of corresponding answers based on given problems, without requiring the model to memorize all information in the samples. In these tasks, the model only needs to learn how to generate correct answers based on input problems, rather than learning the expression of the problems themselves. Therefore, we employ the supervised fine-tuning (SFT) strategy during

training: setting the labels for the problem portions to ignore_index, ensuring these positions about the problems do not participate in loss calculation and gradient updates. This approach allows the model to focus exclusively on learning the mapping relationship from problems to answers.

**Token Permutation (TPM)**    For TPM, **we separately permute the problem and answer components without intermixing them.** Given that the Star Graph and Strongly Connected Components datasets primarily consist of numerical elements, we configure our permutation unit to 2 tokens. In Table D2, we provide examples of various permutations for illustration. Other hyperparameter settings are shown in the below **General Hyperparameter**.

**CPC and CPD**    Consistent with TPM, our permutation unit also consists of 2 tokens. Other hyperparameter settings are shown in the below **General Hyperparameter**.

**General Hyperparameter**    All experiments were conducted on a server equipped with 8 NVIDIA A800 GPUs (80GB each). Training was performed using the `bfloat16` precision format to optimize memory usage and computation. In our experiments, during the SFT phase, we set the maximum sequence length to 128 for Star Graph, with a per-GPU batch size of 64 and a total batch size of 512, full parameters fine-tuning using ZeRO-2 for optimization. Moreover, we set the maximum sequence length to $1,500$ for strongly connected components, with a per-GPU batch size of 64 and a total batch size of 512, full parameters fine-tuning using ZeRO-2 for optimization. We train with bf16 precision, an initial learning rate of $3.0e - 5$, a warm-up ratio of $0.1$, and a cosine scheduler, running for 10 epochs with an early stopping strategy. We use AdamW (Loshchilov & Hutter, 2018) with $\beta_1 = 0.9$, $\beta_2 = 0.95$, and a weight decay of $0.1$.

For **CPC** and **CPD**, the experimental settings are consistent with Appendix D.2.4.

### D.4    POSITIONAL BIAS

#### D.4.1    DATASET INTRODUCTION AND STATISTICS

**Wiki2023+** (Jiang et al., 2024b; Saito et al., 2025) is a real-world benchmark composed of Wikipedia articles published in 2023, selected to minimize overlap with standard LLM pre-training data. To create supervision for question answering, each article is segmented into sentences and individually fed into an LLM to generate QA pairs, with explicit annotations indicating which sentence contains the answer. This sentence-level alignment enables precise analysis of how well models can extract knowledge depending on its position in the training document. Wiki2023+ exhibits natural variability in topic structure, sentence style, and fact density, making it a strong testbed for evaluating model robustness to position and context complexity in real-world settings. The example of Wiki2023+ can be found in the Example D.5.

**Dataset Statistics**    The statistical results of the Wiki2023+ dataset are presented in Table D4.

---

**Example D.5: The example of Wiki2023+**

**Passage** (for continued pre-training):
When Adam Changes (French: Adam change lentement, lit. "Adam Changes Slowly") is a Canadian animated comedy-drama feature film, directed by Joël Vaudreuil and released in 2023. The film centres on Adam, an impressionable teenager growing up in smalltown Quebec who has the unusual quirk that each time somebody makes a comment about his body, whether fair or unfair, his body actually changes to match the comment.
**Question** (for SFT):
When Adam Changes, who directed the Canadian animated comedy-drama feature film?
**Answer**:
Joël Vaudreuil

---

#### D.4.2    BASELINE INTRODUCTION

**AR (Auto-Regressive Training)** is the standard training objective for causal language models. The model is optimized to predict the next token given all previous tokens in the training document. While effective at minimizing perplexity, this approach often results in memorization that is difficult to

| Dataset | Document | Question Answer |
|---------|----------|-----------------|
| Train | 2,385 | 5,493 |
| Test | - | 1,590 |

Table D4: Dataset statistics of Wiki2023+. Since all the documents are seen in the training phase, the number of documents available for testing is "-".

extract through downstream prompts, particularly when the queried information appears in the middle or end of the document.

**Shuffle Sentence** randomly permutes the order of sentences in each training document. This strategy aims to reduce the model's reliance on rigid positional cues and mitigate positional bias. However, disrupting the discourse structure may hinder learning, especially when sentence-level dependencies are important.

**Attn Drop (Attention Dropout)** introduces stochasticity by randomly dropping attention connections during training. This forces the model to depend less on specific token positions, reducing overfitting to earlier context and encouraging more position-invariant representations.

**D-AR (Denoising Auto-Regressive Training)** applies random corruption to a subset of input tokens, replacing them with noise while keeping the output targets unchanged. This method regularizes training by encouraging the model to make robust predictions under partial corruption and has shown the most consistent improvement in extracting knowledge from later document positions.

### D.4.3 EVALUATION METRICS

**EM** metric for this problem is detailed in Appendix D.2.3.

**F1** score is defined as the harmonic mean of precision and recall:

$$\text{F1} = 2 \times \frac{\text{Precision} \times \text{Recall}}{\text{Precision} + \text{Recall}} = \frac{2 \times \text{TP}}{2 \times \text{TP} + \text{FP} + \text{FN}} \tag{16}$$

where **TP** (True Positives) is correctly predicted positive observations; **FP** (False Positives) is incorrectly predicted as positive; **FN** (False Negatives) is incorrectly predicted as negative; **TN** (True Negatives) is correctly predicted negative observations. It is a robust metric that provides a single value to evaluate the performance of the model, especially in scenarios with class imbalance.

### D.4.4 DETAILED IMPLEMENTATION

**CPC and CPD** Our permutation unit consists of 3 words. However, we incorporate the original positional information of permuted words in the original sentence during the forward propagation. Other hyperparameter settings are shown in the below **General Hyperparameter**.

**General Hyperparameter** On the Wiki2023+ dataset, we need to perform continued pre-training to learn the knowledge in the documents, and then perform SFT on the Q&A dataset. Similar to the setting in the Name-description dataset, we treat each document as an independent sample, padding them to the same length using eos_token, while truncating those exceeding the specified length. In our experiments, during the continued pre-training phase, we set the maximum sequence length to $1024$, with a per-GPU batch size of $8$ and a total batch size of $64$, full parameters fine-tuning using ZeRO-2 (Rasley et al., 2020) for optimization. We train with bf16 precision, an initial learning rate of $1.0e - 4$, a warm-up ratio of $0.1$, and a cosine scheduler, running for $150$ epochs with an early stopping strategy. We use AdamW (Loshchilov & Hutter, 2018) with $\beta_1 = 0.9$, $\beta_2 = 0.95$, and a weight decay of $0.1$. During continued pre-training, we evaluate perplexity (PPL) on the training set at each epoch and terminate training early if PPL drops below 2 and the change in PPL between consecutive epochs is $\leq 0.1$.

# E   ANALYSIS & ALATION EXPERIMENTS

## E.1   DISCUSSION: IS IT NORMAL FOR THE SAME PREFIX AND DIFFERENT SUFFIXES?

In this section, we elaborate on the phenomenon of the same prefix and different suffixes. The cause of this phenomenon is that, in the process of permutation learning, it is inevitable to permutate the content order within the sample. Under TPM, the original sentence is often split and recombined. For example, given the sentence "Paul was born on 15 June 1874", token-level permutations may produce sequences such as "Paul was born on June 1874 15" or "Paul was born on 1874 15 June". In this case, the model will train on samples with the same prefix "Paul was born in", but the suffix may differ, such as "June" or "1874". This represents a phenomenon: conflicts in supervisory signals may occur during the model training optimization process, leading to a problem where one suffix probability increases while another decreases. This is a phenomenon that is both normal and abnormal. It is normal because it is produced during the permutation process and is widely present in reality. It is abnormal because it indeed leads to conflicts in the supervisory signals.

During large-scale pre-training on natural corpora, although the phenomenon of "same prefix, different suffix" is commonly observed in real-world language, we argue that such cases should be regarded as independent samples. For example, "I come from city A" and "I come from city B" may both appear in the corpus, but they essentially represent distinct data instances. In other words, A and B indeed each have a 50% probability. In contrast, the samples generated through permutation methods are artificially manipulated from the same underlying data, thereby producing different forms that nevertheless originate from the same semantic content. Therefore, while "same prefix, different suffix" is reasonable in natural corpora, in the context of permutation-based training it does not constitute a new knowledge instance, but rather a perturbation of the same semantic content. Such perturbations no longer provide beneficial diversity, but instead introduce additional learning noise.

## E.2   TRAINING CONVERGENCE ANALYSIS

Figures E1, E2, and E3 illustrate the training convergence curves of four methods (TPM, TPM w/R, CPC, and CPD) across three distinct tasks. Through comparative analysis, we observe that TPM exhibits markedly different convergence characteristics across various task types.

On the name-description dataset (Figure E1), although all methods eventually converge, TPM and TPM w/R (TPM with original relative position) demonstrate significantly slower convergence rates compared to our proposed CPC and CPD. This disparity is particularly evident in the magnified inset, indicating that token permutation methods face optimization challenges even in relatively straightforward text tasks.

However, when transitioning to more complex path planning (Figure E2) and algorithm reasoning tasks (Figure E3), TPM encounters substantially more severe convergence difficulties. In these tasks, the loss reduction for TPM and TPM w/R significantly lags behind CPC and CPD, failing to achieve desirable low loss levels even after extended training periods. Notably, in the algorithm reasoning task, TPM maintains relatively high loss values even after 4,000 training steps.

The fundamental cause of these convergence difficulties can be attributed to the "objective inconsistency" problem induced by token permutation. In TPM, identical input prefixes may correspond to different target outputs because permutations alter the input sequence structure while the expected outputs potentially remain unchanged. This contradiction becomes particularly pronounced in planning and algorithmic reasoning tasks. In contrast, our proposed CPC and CPD methods successfully address this challenge by explicitly modeling positional information. They can identify and process the relationships between permuted tokens and their target positions, thereby ensuring learning consistency while maintaining permutation invariance. This characteristic demonstrates significant advantages across all task types, particularly in planning and algorithmic reasoning tasks that are highly sensitive to sequential order.

## E.3   CAN BIDIRECTIONAL TRAINING ALLEVIATE THE REVERSE CURSE?

In order to verify whether bidirectional training can alleviate the reverse curse, we followed BERT's standard training recipe with MLM as the pre-training task.

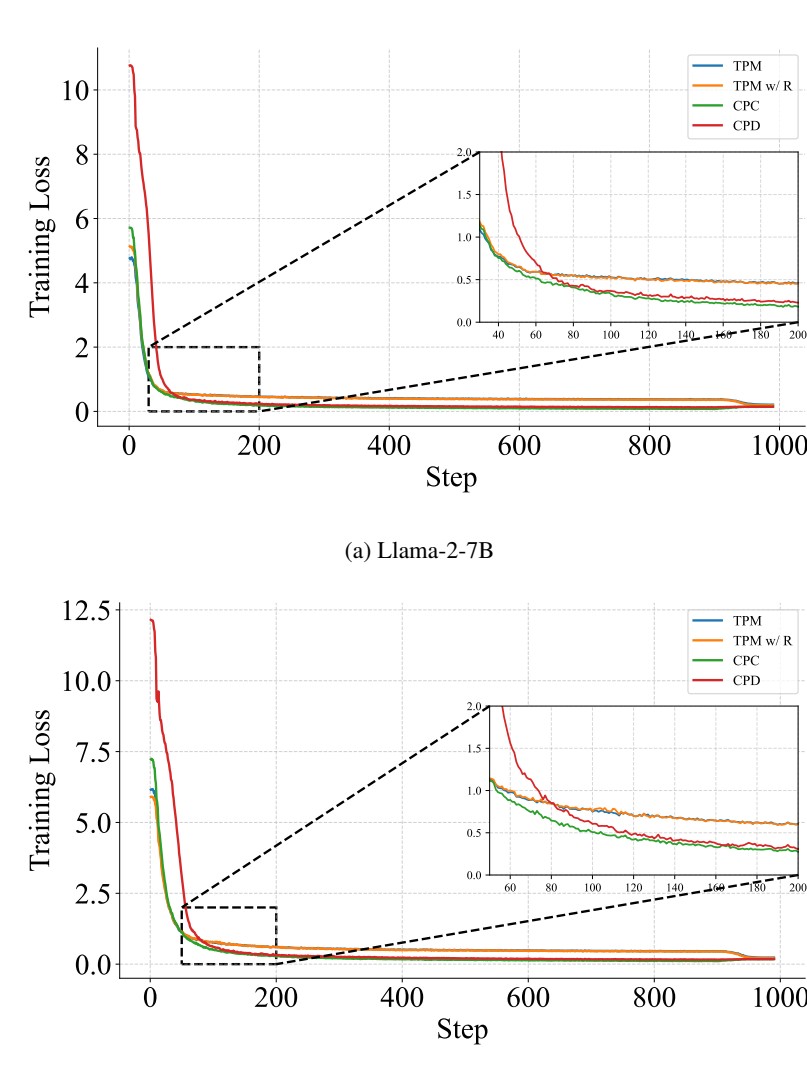

(a) Llama-2-7B

(b) Llama-3.2-1B

Figure E1: Training convergence curve on **name-description** dataset, where `TPM w/R` denotes TPM with token position in original sentence. It can be seen that CPC and CPD have almost the same convergence speed, while TPM and TPM w/R are difficult to converge to the optimum. This difficulty arises mainly from token permutation, which leads to divergent supervision signals under identical prefixes. Such inconsistency is especially detrimental in domains requiring strict sequential dependencies, including path planning and algorithmic reasoning.

**Implementation details**   We use bert-base-uncased (Devlin et al., 2019) for the experiment on the name-description dataset. Each English whole word has a 15% chance of being selected, which is then replaced with a [MASK] token (80% chance), retained (10% chance), or replaced with a random token (10%). Since test set answers may not fall precisely within the 15% masking interval, we experimented with masking rates of 15%, 30%, and 80%. Hyperparameters: max_length=128, batch_size=512 (64*8), learning_rate=8e-5, trained for 100 epochs. During evaluation, consistent with pre-training, we appended the appropriate number of [MASK] tokens to each input based on the expected answer length. We evaluated BERT in two modes: (1) **BERT-parallel**: BERT predicts these masked positions simultaneously; (2) **BERT-AR**: simulating autoregressive generation by predicting tokens sequentially, where each step uses previously generated tokens as context.

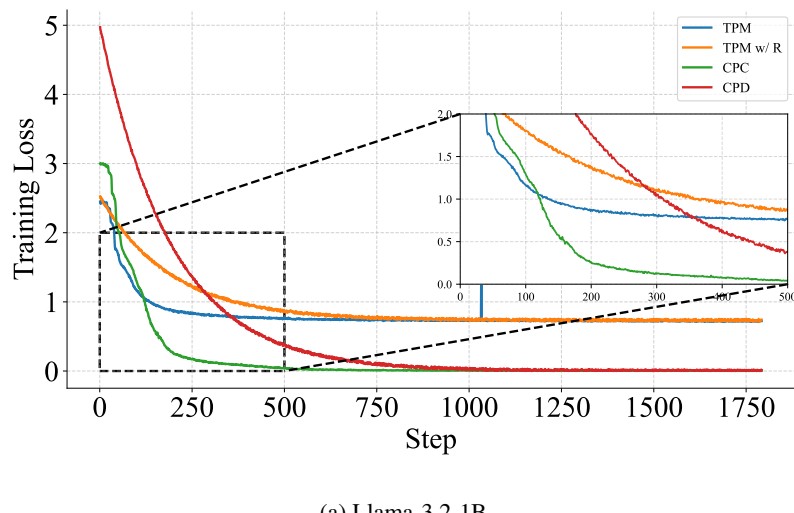

(a) Llama-3.2-1B

Figure E2: Training convergence curve on **path planning** dataset, where `TPM w/ R` denotes TPM with token position in original sentence.

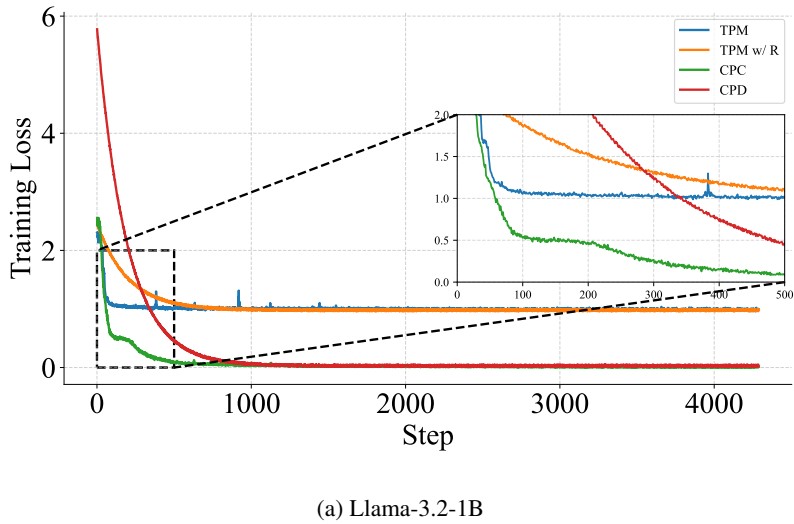

(a) Llama-3.2-1B

Figure E3: Training convergence curve on **algorithm reasoning** dataset, where `TPM w/R` denotes TPM with token position in original sentence.

**Experimental Results** The experimental results are displayed in Table E1. The results reveal several important insights: (1) BERT's bidirectional training struggles with the reversal curse: Despite its bidirectional nature, BERT achieves near-zero exact match scores across all masking rates, with the best performance at 30% masking (9.0% EM) still substantially lower than our methods. (2) Masking rate sensitivity: BERT shows optimal performance at 30% masking, suggesting that neither too sparse (15%) nor too dense (80%) masking effectively captures the required associations for this task. (3) Our methods' superiority: Both CPC and CPD significantly outperform BERT across all metrics, demonstrating that position-aware modeling in autoregressive frameworks is more effective than bidirectional attention for addressing permutation sensitivity.

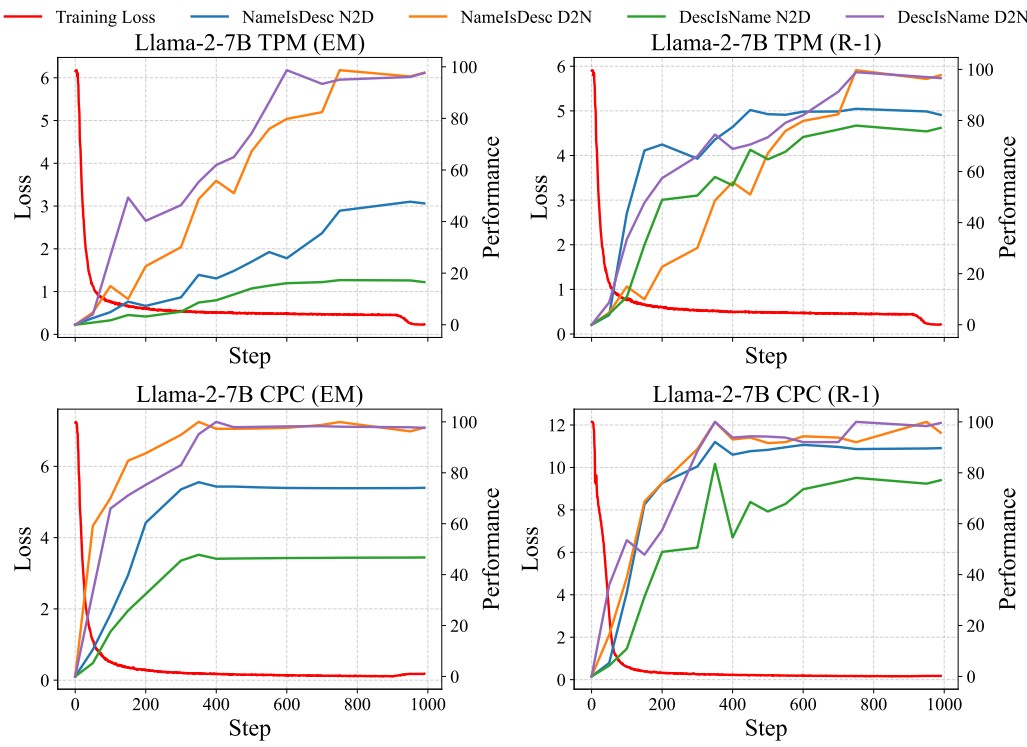

Figure E4: The performance of the Llama-2-7B model changes during the training process. We can find that due to the large number of permutations involved in the TPM training process, exacerbating the conflicting problem of the same prefixes but inconsistent supervision signals. Whereas our proposed CPC introduces position-aware modeling, it can be seen that the convergence is faster and the performance improvement is more obvious.

| Model | N2D in N2D | | | N2D in D2N | | |
|---|---|---|---|---|---|---|
| | **EM** | **R-1** | **BLEU** | **EM** | **R-1** | **BLEU** |
| BERT-parallel (15%) | 0.0 | 12.2 | 15.8 | 0.0 | 13.0 | 17.4 |
| BERT-parallel (30%) | 9.0 | 22.4 | 28.1 | 0.3 | 15.8 | 22.7 |
| BERT-parallel (80%) | 0.0 | 11.8 | 15.9 | 0.0 | 12.8 | 17.5 |
| BERT-AR (15%) | 0.0 | 12.0 | 15.3 | 1.0 | 13.7 | 17.9 |
| BERT-AR (30%) | 2.3 | 14.3 | 18.0 | 2.0 | 14.8 | 19.6 |
| BERT-AR (80%) | 0.0 | 12.2 | 15.7 | 0.0 | 12.9 | 17.4 |
| Llama-2-7B-CPC | 76.2±0.2 | 91.8±0.8 | 93.2±0.4 | 47.5±0.3 | 83.2±0.6 | 92.0±0.4 |
| Llama-2-7B-CPD-6L | 78.1±0.4 | 92.2±0.5 | 94.2±0.6 | 47.9±0.7 | 85.4±0.5 | 93.7±0.4 |
| Llama-3.2-1B-CPC | 78.6±0.2 | 91.5±0.4 | 92.3±0.2 | 32.6±0.3 | 82.5±0.7 | 89.5±0.3 |
| Llama-3.2-1B-CPD-6L | 81.5±0.4 | 94.0±1.2 | 95.6±0.5 | 62.7±0.5 | 84.9±0.7 | 87.2±0.9 |

Table E1: Comparison with the bidirectional training model BERT. To eliminate the problem of random error, we conducted five seed experiments on CPC and CPD, and the experimental results are expressed as **mean ± standard** deviation.

### E.4 DOES CPC&CPD TRAINING HURT PERFORMANCE ON STANDARD TASKS?

In our main experiments, we demonstrated that CPC and CPD achieve promising performance on three common failure modes of NTP. A natural concern, however, is that **since the pre-training phase does not involve any position-aware training objectives, extensive permutation-based training might risk overfitting to these benchmark datasets of failure modes, potentially leading to catastrophic**

**forgetting**. To address this concern, we further investigate whether CPC and CPD disrupt zero-shot performance on eight standard evaluation tasks, including BoolQ (Clark et al., 2019), PIQA (Bisk et al., 2020), SIQA (Sap et al., 2019), HellaSwag (Zellers et al., 2019), WinoGrande (Sakaguchi et al., 2020), ARC (easy and challenge) (Clark et al., 2018), OpenBookQA (Mihaylov et al., 2018), and 5-shot aggregated MMLU (Hendrycks et al., 2020) dataset.

### E.4.1 DATASET INTRODUCTION AND STATISTICS

In this section, we introduce the datasets used to evaluate the LLMs' zero-shot and 5-shot performance, along with the prompt examples employed in the evaluation. We also present the corresponding dataset statistics in Table E2.

- **BoolQ**[5] dataset is specifically designed for yes/no question answering tasks. Unlike artificially constructed queries, the questions in BoolQ originate from naturally occurring real-world scenarios, characterized by spontaneity and openness. Each instance in the dataset consists of three components: a question, a corresponding passage, and an answer. In terms of task formulation, the model is presented with a passage and required to answer the given question based on that passage, with the answer constrained to either **True** or **False**. Since the test set does not have public answers, we use the validation set for evaluation.

> **Example E.1: The prompt of BoolQ**
>
> **instruction**:
> Please answer the given 'Question' based on the following 'Passage', and only respond with 'True' or 'False'.
> **input**:
> Passage:
> In mathematics, parity is the property of an integer's inclusion in one of two categories: even or odd. An integer is even if it is evenly divisible by two and odd if it is not even. For example, 6 is even because there is no remainder when dividing it by 2. By contrast, 3, 5, 7, 21 leave a remainder of 1 when divided by 2. Examples of even numbers include −4, 0, 82 and 178. In particular, zero is an even number. Some examples of odd numbers are −5, 3, 29, and 73.
> Question:
> can an odd number be divided by an even number?
> **Answer**:

- **PIQA**[6] dataset is for physical commonsense reasoning. It contains questions about everyday scenarios that require practical knowledge of physical interactions, with answers often favoring unconventional but plausible solutions. In terms of task formulation, PIQA provides a context about a physical situation, and the model is required to choose the correct answer between two candidate solutions (**A** or **B**), where only one reflects valid physical commonsense.

> **Example E.2: The prompt of PIQA**
>
> **instruction**:
> Please determine which of the two answers is more accurate and helpful for the following question. You must answer with either 'A' or 'B' only.
> **input**:
> Question:
> dresser
> A. replace drawer with bobby pin
> B. finish, woodgrain with bobby pin
> **Answer**:

- **SIQA**[7](Social IQa) is a benchmark for social commonsense reasoning. Unlike datasets focused on physical or taxonomic knowledge, it centers on understanding people's actions

---

[5]https://huggingface.co/datasets/google/boolq
[6]https://huggingface.co/datasets/ybisk/piqa
[7]https://huggingface.co/datasets/allenai/social_i_qa

and their social implications. Each instance presents an action and a question with multiple candidate answers (**A**, **B** or **C**), only one of which reflects plausible social reasoning.

---

**Example E.3: The prompt of SIQA**

**instruction**:
You are given a situation, a question, and three possible answers. Choose the best answer that most reasonably and socially fits the situation.
**input**:
Context:
Sasha protected the patients' rights by making new laws regarding cancer drug trials.
Question:
What will patients want to do next?
A. write new laws
B. get petitions signed
C. live longer
Please respond with only the letter of the best answer (A, B, or C).
**Answer**:

---

- **HellaS**[8] dataset is for commonsense natural language inference, specifically targeting the ability of models to select the most plausible continuation of a given context. Each instance presents a short context and four candidate endings (**A**, **B**, **C**, or **D**), only one of which is correct.

---

**Example E.4: The prompt of HellaS**

**instruction**:
You are given a context and four possible endings. Choose the best ending that most reasonably and logically completes the context.
**input**:
Context:
A boy is running down a track. the boy
A. runs into a car.
B. gets in a mat.
C. lifts his body above the height of a pole.
D. stands on his hands and springs.
Please respond with only the letter of the best answer (A, B, C, or D).
**Answer**:

---

- **WinoG**[9] dataset is a commonsense reasoning benchmark inspired by the Winograd Schema Challenge, designed to address its limitations in scale and dataset-specific bias. Each instance presents a sentence with a blank and two candidate options (**A** or **B**), only one of which is correct.

---

**Example E.5: The prompt of WinoG**

**instruction**:
You are given a sentence with a blank (_) and two possible options. Choose the option that best and most logically fills in the blank.
**input**:
Sentence:
The doctor diagnosed Justin with bipolar and Robert with anxiety. _ had terrible nerves recently.
A. Justin
B. Robert
Please respond with only the letter of the best answer (A or B).
**Answer**:

---

- **ARCe** and **ARCc**[10] are two subsets of the AI2 Reasoning Challenge, a benchmark of grade-school science questions. The **Easy Set** (ARCe) contains questions solvable by

---

[8]https://huggingface.co/datasets/Rowan/hellaswag
[9]https://huggingface.co/datasets/allenai/winogrande
[10]https://huggingface.co/datasets/allenai/ai2_arc

simple retrieval or co-occurrence methods, whereas the **Challenge Set** (ARCc) consists of questions that these methods fail to answer, thus requiring deeper reasoning. Each instance is a multiple-choice question with four options (**A**, **B**, **C**, or **D**), only one of which is correct.

---

**Example E.6: The prompt of ARCe and ARCc**

**instruction**:
You are given a multiple-choice science question. Choose the best answer based on reasoning and knowledge.
**input**:
Question:
An astronomer observes that a planet rotates faster after a meteorite impact. Which is the most likely effect of this increase in rotation?
A. Planetary density will decrease.
B. Planetary years will become longer.
C. Planetary days will become shorter.
D. Planetary gravity will become stronger.
Please respond with only the letter of the best answer (A, B, C, or D).
**Answer**:

---

- **OBQA**[11] dataset is specifically designed to evaluate advanced question-answering abilities. Unlike simple fact-recall tasks, the questions in OpenBookQA require multi-step reasoning and the integration of both scientific knowledge and common sense. Each instance consists of a science question, several answer choices (**A**, **B**, **C**, or **D**), and access to a set of core science facts (the "open book") provided with the dataset.

---

**Example E.7: The prompt of OBQA**

**instruction**:
You are given a multiple-choice science question. Choose the best answer based on reasoning and knowledge.
**input**:
Question:
Predators eat
A. lions
B. humans
C. bunnies
D. grass
Please respond with only the letter of the best answer (A, B, C, or D).
**Answer**:

---

- **MMLU**[12] is a benchmark for evaluating multitask language understanding across a wide range of academic subjects. Each instance is a multiple-choice question with four candidate answers (**A**, **B**, **C**, or **D**), where the model must identify the correct option by combining world knowledge with reasoning ability. Given the difficulty and diversity of tasks, we randomly sample five validation examples of the same type as few-shot demonstrations when evaluating on the test set.

---

[11]https://huggingface.co/datasets/allenai/openbookqa
[12]https://huggingface.co/datasets/cais/mmlu

| Dataset | BoolQ | PIQA | SIQA | HellaS | WinoG | ARCe | ARCc | OBQA | MMLU |
|---|---|---|---|---|---|---|---|---|---|
| Eval Number | 3,270 | 1,838 | 1,954 | 10,042 | 9,248 | 2,376 | 1,172 | 500 | 14,042 |

Table E2: Statistics of nine traditional natural language processing evaluation benchmarks.

| | Methods | BoolQ | PIQA | SIQA | HellaS | WinoG | ARCe | ARCc | OBQA | MMLU | Avg |
|---|---|---|---|---|---|---|---|---|---|---|---|
| Original | | 63.9 | 45.5 | 32.9 | 25.0 | 50.3 | 24.0 | 22.4 | 27.6 | 24.2 | **35.5** |
| NTP | Standard | 47.3 | 49.5 | 33.5 | 25.0 | 52.3 | 24.8 | 22.5 | 28.0 | 24.3 | 33.9 |
| | w PAE | 9.5 | 49.5 | 32.9 | 25.0 | 50.3 | 23.9 | 22.4 | 27.6 | 23.5 | 29.4 |
| TPM | Standard | 0.0 | 49.4 | 32.7 | 25.0 | 49.7 | 24.3 | 22.1 | 25.6 | 24.3 | 28.1 |
| CPC | 🔥 All | 4.2 | 48.5 | 33.6 | 24.6 | 49.6 | 22.7 | 23.5 | 22.0 | 25.4 | 28.2 |
| | 🔥 Embedding | 0.0 | 48.4 | 32.9 | 24.5 | 49.8 | 21.6 | 23.8 | 22.0 | 24.4 | 27.5 |
| | 🔥 Transformers | 8.8 | 49.2 | 32.5 | 25.3 | 50.6 | 24.1 | 24.6 | 21.6 | 23.6 | 28.9 |
| CPD-6L | 🔥 All | 0.0 | 48.6 | 33.1 | 24.5 | 50.3 | 24.0 | 23.0 | 25.6 | 24.2 | 28.1 |
| | 🔥 Transformers | 26.0 | 48.0 | 32.1 | 24.7 | 50.0 | 23.0 | 23.4 | 25.2 | 25.0 | 30.8 |
| | 🔥 Transformers (14-15) | 48.6 | 49.5 | 33.6 | 25.0 | 50.5 | 24.1 | 22.6 | 27.6 | 25.6 | 34.1 |
| | 🔥 Transformers (13-15) | 54.0 | 49.5 | 33.6 | 25.0 | 50.1 | 25.4 | 22.4 | 28.0 | 25.9 | 34.9 |
| | 🔥 Transformers (12-15) | 52.7 | 49.5 | 33.6 | 25.0 | 50.3 | 24.2 | 22.4 | 27.8 | 25.2 | 34.5 |
| | 🔥 Transformers (11-15) | 55.2 | 49.5 | 33.6 | 25.0 | 48.7 | 26.6 | 24.2 | 27.6 | 25.3 | 35.1 |
| | 🔥 Transformers (10-15) | 54.2 | 49.7 | 33.8 | 25.1 | 50.2 | 24.2 | 23.5 | 27.2 | 25.1 | 34.8 |
| | 🔥 Transformers (9-15) | 30.2 | 49.5 | 33.5 | 25.2 | 52.1 | 27.2 | 24.2 | 26.8 | 24.9 | 32.6 |

Table E3: Performance results of various fine-tuned versions of Llama-3.2-1B on standard benchmarks. Here, we investigate which part of the fine-tuned parameters has an impact on the original LLMs' ability. Original denotes the base model. All other models are fine-tuned on the name-to-description dataset. w/ PAE indicates the position-aware embedding introduced during fine-tuning. The 🔥 XX signifies that only the parameters of component XX in the base model are trained. Transformers $(i–j)$ refers to fine-tuning all Transformer blocks from layer $i$ to layer $j$. If no specific range is indicated, the fine-tuning is applied to all Transformer layers.

---

**Example E.8: The prompt of MMLU**

**instruction**:
The following are multiple choice questions (with answers) about {task type}.
**input**:
Question:
Same type of task question 1, answer choice, and the corresponding answer.
Same type of task question 2, answer choice, and the corresponding answer.
Same type of task question 3, answer choice, and the corresponding answer.
Same type of task question 4, answer choice, and the corresponding answer.
Same type of task question 5, answer choice, and the corresponding answer.
current question and answer choice.
**Answer**:

---

### E.4.2 IMPLEMENTATION DETAILS & EXPERIMENTAL RESULTS

**Implementation details** The proposed position-aware modeling is primarily designed to mitigate common failure modes of standard NTP, rather than to pre-train a LLM from scratch (which we leave for future work). Therefore, when evaluating whether the general performance is affected, we remove the position-aware modules at the testing stage, namely the position embeddings in CPC and the position-aware block layers in CPD. Specifically, for fine-tuned models, NTP and TPM introduce no additional components and can thus be directly evaluated with the fine-tuned model. For NTP (w/ PAE), the position-aware embeddings are incorporated during training but removed during evaluation. Similarly, for CPC and CPD variants, we retain only the original fine-tuned base model structure during evaluation, while the additional position-aware components are excluded.

| Method | NameIsDescription | | | | | DescriptionIsName | | | | |
|---|---|---|---|---|---|---|---|---|---|---|
| | N2D | | | D2N | | N2D | | | D2N | |
| | EM | R-1 | BLEU | EM | R-1 | EM | R-1 | BLEU | EM | R-1 |
| Llama-3.2-1B-base | | | | | | | | | | |
| CPD-6L 🔥 Transformers (14-15) | 62.7 | 74.9 | 77.4 | 93.3 | 93.3 | 49.7 | 66.1 | 69.3 | **100.0** | **100.0** |
| 🔥 Transformers (13-15) | 63.7 | 76.0 | 78.4 | 96.0 | 96.0 | 53.0 | 69.0 | 72.0 | **100.0** | **100.0** |
| 🔥 Transformers (12-15) | 64.0 | 76.3 | 78.9 | 96.3 | 96.3 | 53.3 | 79.8 | 72.9 | 99.0 | 99.0 |
| 🔥 Transformers (11-15) | 65.3 | 77.5 | 79.9 | 99.7 | 99.7 | 54.7 | 71.7 | 74.7 | 99.3 | 99.3 |
| 🔥 Transformers (10-15) | 66.0 | 78.2 | 80.7 | 98.3 | 98.3 | 58.9 | 75.0 | 77.7 | 99.7 | 99.7 |
| 🔥 Transformers (9-15) | 70.3 | 82.4 | 84.5 | **100.0** | **100.0** | 59.0 | 75.2 | 77.9 | **100.0** | **100.0** |
| 🔥 All | **81.3** | **94.7** | **95.8** | **100.0** | **100.0** | **63.0** | **85.3** | **87.7** | **100.0** | **100.0** |

Table E4: Performance of the CPD variant on the name-description dataset. Complementary to Table E3, the performance of downstream tasks needs to be guaranteed while retaining the performance of the original model.

**Experimental Results** The experimental results are summarized in Table E3 and Table E4, from which we draw the following conclusions:

(1) **Universality and controllability of catastrophic forgetting.** Compared with the performance of the original model (35.5% on average), even standard NTP substantially degrades the general capabilities of the model (33.9% on average), indicating that catastrophic forgetting is a widespread issue. However, our CPD method can effectively mitigate this phenomenon by precisely controlling the degree of base model freezing. Specifically, for the Llama-3.2-1B model with 16 Transformer layers, when fine-tuning only the top few layers (*e.g.*, CPD-Transformers 11–15), the average performance drops by merely 0.4% (from 35.5% to 35.1%), demonstrating the effectiveness of our approach in preserving the model's original capabilities.

(2) **Impact of coupling vs. decoupling content and position.** CPC introduces position-awareness by directly adding positional embeddings to the original input embeddings. This tight coupling of content and positional information leads to semantic drift in the learned representations. As a result, different CPC configurations (All: 28.2%, Embedding: 27.5%, Transformers: 28.9%) all perform significantly worse than the original model, **underscoring the negative impact of inconsistent paradigms between pre-training and fine-tuning**. In contrast, CPD achieves a modular decoupling of content and positional information through dedicated position-aware blocks, while preserving the structural integrity of the base model. When fine-tuning only a subset of Transformer layers (*e.g.*, CPD-Transformers 11–15: 35.1%), the performance remains nearly identical to that of the original model, validating the advantage of the decoupled design.

(3) **Layer sensitivity and trade-offs in fine-tuning strategies.** The results reveal a trade-off between adapting to new tasks and retaining pre-trained knowledge. When all base model parameters are fine-tuned (CPD-All: 28.1%), the model achieves the best performance on position-aware tasks but suffers from a sharp decline in general capabilities due to extensive parameter changes. Interestingly, as more layers are fine-tuned, we observe an improvement rather than a degradation: performance rises from 34.1% with CPD-Transformers (14–15) to 35.1% with CPD-Transformers (11–15). This suggests that moderate parameter fine-tuning, coupled with permutation-invariant training, allows the model to retain pre-trained knowledge while gaining additional position-aware abilities.

(4) **Task-specific performance preservation.** Table E4 provides deeper insights into how our method maintains performance on the target position-aware tasks while preserving general capabilities. Notably, most CPD configurations show strong performance on the challenging name-description tasks, demonstrating robust position-invariant learning. The CPD-Transformers (11–15) configuration achieve an optimal balance, maintaining strong performance on both forward (N2D: 65.3% EM) and reverse (D2N: 99.7% EM) name-description tasks while achieving the best preservation of general capabilities (35.1% average). This verifies that our framework can both endow the model

| Method | Parameter | NameIsDescription | | | | | DescriptionIsName | | | | |
|---|---|---|---|---|---|---|---|---|---|---|---|
| | | N2D | | | D2N | | N2D | | | D2N | |
| | | EM | R-1 | BLEU | EM | R-1 | EM | R-1 | BLEU | EM | R-1 |
| | | | | | Llama-2-7B-base | | | | | | |
| CPC | 6.74B | 76.3 | 92.1 | 93.1 | **100.0** | **100.0** | 47.8 | 83.5 | 92.3 | **100.0** | **100.0** |
| 1-L | 7.07B | 76.5 | 89.3 | 93.5 | **100.0** | **100.0** | 46.5 | 84.9 | 92.3 | **100.0** | **100.0** |
| 3-L | 7.41B | 77.2 | 91.3 | 93.2 | 98.3 | 98.3 | 47.9 | 84.2 | 92.8 | 99.7 | 99.7 |
| CPD 6-L | 7.92B | 78.3 | 91.9 | 94.4 | **100.0** | **100.0** | 48.3 | 85.7 | 93.6 | **100.0** | **100.0** |
| 8-L | 8.25B | 79.2 | 92.5 | 95.0 | **100.0** | **100.0** | 47.6 | 84.3 | 92.8 | **100.0** | **100.0** |
| 12-L | 8.93B | **79.9** | **93.7** | **96.2** | **100.0** | **100.0** | **52.6** | **87.3** | **95.2** | **100.0** | **100.0** |
| | | | | | Llama-3.2-1B-base | | | | | | |
| CPC | 1.23B | 78.7 | 91.8 | 92.8 | 82.7 | 83.6 | 32.8 | 82.9 | 89.7 | **100.0** | **100.0** |
| 1-L | 1.57B | 79.6 | 91.9 | 93.0 | 86.7 | 86.7 | 31.5 | 83.0 | 88.6 | **100.0** | **100.0** |
| 3-L | 1.68B | 80.5 | 92.2 | 93.7 | 99.6 | 99.6 | 43.7 | 83.8 | 87.9 | **100.0** | **100.0** |
| CPD 6-L | 1.86B | 81.3 | 94.7 | 95.8 | **100.0** | **100.0** | 63.0 | 85.3 | 87.7 | **100.0** | **100.0** |
| 8-L | 1.98B | 81.9 | 95.3 | 96.2 | **100.0** | **100.0** | 63.4 | 85.8 | 88.1 | **100.0** | **100.0** |
| 12-L | 2.21B | **82.8** | **95.9** | **96.3** | **100.0** | **100.0** | **65.8** | **87.2** | **90.1** | **100.0** | **100.0** |

Table E5: Experimental results on the reversal curse setting. $i$-L denotes the number of position-aware layers, with CPD (6-L) serving as the default configuration throughout all experiments.

with permutation invariance and maintain the model's generalization ability, preventing excessive catastrophic forgetting from occurring.

## E.5 Ablation Experiment

### E.5.1 The number of position-aware blocks

We conduct comprehensive ablation experiments to investigate the impact of the number of position-aware blocks on model performance in the reversal curse setting. As shown in Table E5, we evaluate CPD architectures with varying numbers of position-aware layers on NameIsDescription (N2D) and DescriptionIsName (D2N) tasks using two base models: Llama-2-7B and Llama-3.2-1B.

Our results reveal several key findings: (1) CPD consistently achieves perfect or near-perfect performance (EM scores of 100.0) on the reversed D2N task across most layer configurations, demonstrating their effectiveness in handling permutation-invariant tasks. (2) We observe a general trend of performance improvement as the number of position-aware layers increases, with the 6-L configuration emerging as an optimal balance between performance and parameter efficiency. For instance, in the Llama-2-7B CPD model, BLEU scores on N2D improve from 91.3 (3-L) to 91.9 (6-L), while maintaining perfect scores on D2N tasks.

Notably, the performance gains begin to plateau beyond 6 layers, with diminishing returns observed in the 8-L and 12-L configurations. This suggests that 6 position-aware layers provide sufficient capacity to capture the necessary positional relationships for effective permutation-invariant learning. The consistent superiority of the 6-L configuration across both model sizes and task directions validates our choice of CPD (6-L) as the default setting throughout our experiments.

| Method | Parameter | NameIsDescription | | | | | DescriptionIsName | | | | |
|---|---|---|---|---|---|---|---|---|---|---|---|
| | | N2D | | | D2N | | N2D | | | D2N | |
| | | EM | R-1 | BLEU | EM | R-1 | EM | R-1 | BLEU | EM | R-1 |
| CPC | 6.74B | 76.3 | 92.1 | 93.1 | 100.0 | 100.0 | 47.8 | 83.5 | 92.3 | 100.0 | 100.0 |
| CPD 12-L | 8.93B | **79.9** | **93.7** | **96.2** | 100.0 | 100.0 | **52.6** | **87.3** | **95.2** | 100.0 | 100.0 |
| CPD Frozen ALL | 2.32B | 48.7 | 72.4 | 76.3 | 28.3 | 29.6 | 3.3 | 27.8 | 33.0 | 99.7 | 99.7 |
| CPD Frozen Embedding | 8.93B | 73.0 | 87.9 | 90.9 | 98.3 | 98.3 | 47.3 | 75.7 | 79.6 | 99.0 | 99.0 |

Table E6: Experimental results on the reversal curse setting with Llama-2-7B. $i$-L denotes the number of position-aware layers, **Frozen ALL** means freeze all parameters of the pre-trained AR models, and **Frozen Embedding** represents only freezing the parameters of the embedding layer in the pre-trained AR models.

### E.5.2 WHETHER TO TRAIN THE PRE-TRAINED AR MODELS IN CPD

In CPD, we append multiple layers of our proposed position-aware blocks after the output layer of the existing pre-trained AR models, effectively decoupling the target position and content representations, with target positions serving as query vectors. A natural question arises: can we train only the position-aware blocks while keeping the parameters of the pre-trained AR models fixed? To investigate this, we conducted comparative experiments on the name-description dataset using Llama-2-7B, with results presented in Table E6. The following conclusions can be drawn: (1) **Frozen ALL** (training only position-aware blocks while completely freezing pre-trained AR models parameters) demonstrates significantly degraded performance. On the NameIsDescription N2D task, performance drops precipitously from 79.9 (EM) and 93.7 (R-1) for CPD-12L to 48.7 (EM) and 72.4 (R-1). More severely, on the DescriptionIsName N2D task, performance almost completely collapses, declining from 52.6 (EM) and 87.3 (R-1) to merely 3.3 (EM) and 27.8 (R-1). This substantial performance deterioration primarily occurs because knowledge-related content representations are predominantly stored within the pre-trained AR models. When these parameters are frozen, the model cannot adjust its internal knowledge representations to accommodate the position-aware mechanism. Although position-aware blocks can theoretically store some knowledge information, their design primarily focuses on processing positional information rather than content representation, resulting in limited knowledge storage capacity.

(2) In contrast, **Frozen Embedding** (freezing only the embedding layer while allowing updates to other parameters) exhibits performance more closely approximating the fully fine-tuned model. On the NameIsDescription task, this strategy achieves 73.0 (EM) and 87.9 (R-1), which, while slightly lower than the fully fine-tuned CPD-12L, significantly outperforms the **Frozen ALL**. On the DescriptionIsName task, **Frozen Embedding** approaches the performance of the fully fine-tuned model, with nearly identical results on the D2N task (98.3 vs. 100.0).

These results indicate that updating pre-trained AR models parameters (particularly parameters beyond the embedding layer) during training is crucial for effectively integrating positional information and content representations.

### E.5.3 THE UNIT OF PERMUTATION

To confirm the impact of permutation unit granularity on model performance, we conducted experiments on permutation unit granularity under the reversal curse setting, and the results are shown in Figure E5. We can draw the following conclusions: (1) Both small and large permutation units are detrimental to model performance. When permutation units are too small (*e.g.*, 1-2 words), the model is forced to learn fragmented representations of common linguistic phrases and fixed collocations, which imposes an additional learning burden and disrupts the natural semantic coherence of language constructs. Conversely, when permutation units are too large (*e.g.*, 7+ words), the model cannot effectively perceive and adapt to different degrees of contextual variations, as the permutation granularity becomes too coarse to provide meaningful positional diversity during training. (2) The results reveal that different task exhibit distinct optimal permutation unit sizes. For the N2D task within

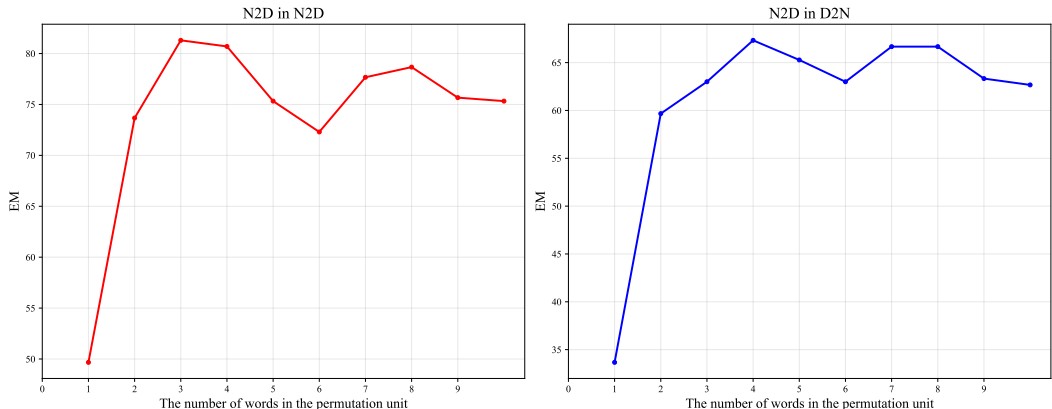

Figure E5: Effect of permutation unit size on reversal curse performance using Llama-3.2-1B-CPD (6-L). EM scores are shown for different word-level permutation unit sizes on the name-description dataset. Left: N2D task performance in NameIsDescription setting. Right: N2D task performance in DescriptionIsName setting.

the NameIsDescription setting, peak performance is achieved around 3-4 words per permutation unit, while the N2D task within the DescriptionIsName setting shows optimal performance around 4-5 words per unit. This suggests that the complexity and structure of the underlying task influence the most effective permutation granularity. (3) The consistent decline in performance at both extremes suggests that maintaining an appropriate balance between providing positional diversity and preserving semantic coherence is essential for effective permutation-based training.

