# OpenReview forum: "Position-Aware Modeling for Next-Token Prediction"
_ICLR.cc/2026/Conference — Submitted to ICLR 2026_

### Official Review · Reviewer_nPtM · 2025-10-28

**Soundness:** 1
**Presentation:** 1
**Contribution:** 2
**Rating:** 2
**Confidence:** 3

**Summary:**

This paper identifies the inherent sensitivity to token ordering in standard Next-Token Prediction (NTP) training as a root cause for several failure modes in LLMs, such as impaired planning and reasoning. The authors argue this sensitivity leads to issues like the reversal curse and knowledge position sensitivity. To address this, they propose a "position-aware enhancement framework." This framework is designed to improve standard training by enabling the model to distinguish prediction targets based on both their content and their intended output position. The authors claim that this method significantly improves the model's robustness to variations in token order and allows smaller models to outperform larger-scale LMs on certain tasks.

**Strengths:**

* The authors correctly identify that the strict left-to-right, autoregressive objective is linked to several well-documented failure modes in LLMs, including the reversal curse, the factorization curse, and sensitivity to the position of knowledge in the context.
* Addressing this fundamental issue is a valuable research direction, and a successful solution would represent an important contribution to the field.

**Weaknesses:**

The paper's central premise is to overcome the limitations of fixed left-to-right generation. However, it fails to cite or compare against a large and directly relevant body of work on any-order autoregressive models [1, 2, 3]. These methods are explicitly designed to break the left-to-right dependency structure in LLMs.

In general, the paper is poorly written and difficult to understand. The description of the method is vague and imprecise.

- The paper claims "teacher-forcing... is a problem in NTP" (lines 68-69) and defines a standard MLE objective in Eq. 2, but it never clearly explains why teacher-forcing is the problem it aims to solve, nor how its method (which also appears to be a supervised objective) avoids this supposed problem.

- The authors use highly informal language, such as the "downside of potentially messing with the original token embedding" (line 109), which is inappropriate for a scientific paper.

- The paper introduces concepts like "semantic representations learned during pre-training to drift" (lines 269-271) without providing a clear definition of what this "drift" is, why it's a problem, or how the proposed 'CPD' mechanism resolves it.

- Equation 3 is referred to as an "objective" (line 172), but it does not appear to be an objective function to be optimized.

These examples are not exhaustive; the entire method section is difficult to parse, and reading it carefully does not resolve the ambiguity.

[1] Hoogeboom, Emiel, et al. "Autoregressive diffusion models." arXiv preprint arXiv:2110.02037 (2021).

[2] Shih, Andy, Dorsa Sadigh, and Stefano Ermon. "Training and inference on any-order autoregressive models the right way." Advances in Neural Information Processing Systems 35 (2022): 2762-2775.

[3] Pannatier, Arnaud, Evann Courdier, and François Fleuret. "σ-gpts: A new approach to autoregressive models." Joint European Conference on Machine Learning and Knowledge Discovery in Databases. Cham: Springer Nature Switzerland, 2024.

**Questions:**

1. The paper states (lines 313-315) that the attention mask "retains [the] left-to-right generation constraint in vanilla NTP." If generation at inference time is still strictly left-to-right, what is the effect of your position-aware training? How does the model's behavior at inference time differ from a standard autoregressive model?

2. The proposed method appears to add extra parameters to the model. Is it possible that any observed performance gains are simply due to an increase in model capacity? Please provide a precise breakdown of the parameter overhead introduced by your method and include comparisons to baselines with a similar total parameter count.

3. The authors use the term "permutation-invariant language modeling" (line 485). Language is inherently sequential and not permutation-invariant (e.g., "dog bites man" vs. "man bites dog"). What is meant by "permutation-invariance" in this context?

---

> ### Author Response · Authors · 2025-11-21
> **Response to Reviewer nPtM (PART I)**
>
> Thank you very much for taking the time to review and for your support. We try our best to address your questions as follows.
>
> > The paper's central premise is to overcome the limitations of fixed left-to-right generation. However, it fails to cite or compare against a large and directly relevant body of work on any-order autoregressive models [1, 2, 3]. These methods are explicitly designed to break the left-to-right dependency structure in LLMs.
>
> We sincerely appreciate the reviewer for pointing out these related works. In the revised version, we have included citations to these necessary papers and provided corresponding discussions in the **Related Work** section. **Our core contribution is fundamentally different from the problems addressed by existing any‑order autoregressive models**.
>
> 1. **Scope**: The scope of our research differs from that of ARDMs, AO-ARMs, and σ‑GPTs.
>
>     - First, we would like to clarify that ARDMs and AO‑ARMs **do not fall within the category of mainstream AR language models**. ARDMs are essentially discrete diffusion models, employing an absorbing diffusion process and an ELBO training objective. During inference, it requires multi‑step iterative sampling, similar to image diffusion models. AO‑ARMs, on the other hand, require training $N2^{N-1}$ conditional distributions and adopt a specialized absorbing‑state token mechanism. Both approaches represent fundamentally different paradigms.
>
>     - In contrast, σ‑GPTs are the most similar to our work among the three, as they also perform AR modeling. Nevertheless, our work differs substantially in both **motivation** and **Method**, as detailed in the following parts.
> 2. **Motivation**: In contrast to prior studies that aim to establish entirely new generative paradigms, our work focuses on **refining the training paradigm** of existing models, while **preserving the standard left-to-right generation process**.
>
>     - The objective of σ-GPTs, ARDMs, and AO-ARMs is to achieve **arbitrary-order generation**, which is particularly applicable to data lacking an intrinsic sequential structure (*e.g.* , images) or to tasks demanding flexible conditional generation, such as infilling and image inpainting.
>     - Our study (1) attributes three common failure modes observed in **mainstream pre-trained LLMs** to the lack of permutation invariance induced by vanilla left-to-right NTP, rather than to the "generation order constraint"; and (2) propose a position-aware modeling approach that endows the model with permutation invariance, enabling it to learn similar joint probability distributions for different permutations of the same sample **during training**, without altering the model’s original left-to-right generation process.
> 3. **Method Comparison with** **$\sigma$** **-GPTs:**  Beyond motivation, our approach offers distinct advantages over $\sigma$-GPTs, particularly regarding compatibility with modern LLM architectures.
>
>     - **Architectural Incompatibility of** **$\sigma$** **-GPTs:**  $\sigma$-GPTs require learning three distinct embeddings (token, position, target position) for every token and concatenating them at the **input layer**. This design faces a critical limitation: it is fundamentally incompatible with the relative position encodings (*e.g.* , **RoPE, ALiBi**) that dominate modern LLMs.
>     - **The Advantage of CPC:**  In contrast, CPC offers a lightweight and compatible solution.
>
>       - **Mechanism:**  Instead of maintaining $T$ separate target embeddings, CPC learns a single shared **position-aware embedding**. It leverages the RoPE mechanism to dynamically generate representations for different target positions.
>       - **Efficiency:**  This design requires only one parameters and is agnostic to the underlying architecture, allowing seamless integration into any model using RoPE, ALiBi, or absolute encodings.
>     - **The Advantage of CPD:**  CPD takes this further by **explicitly decoupling** content and target position. By handling target position information in separate position-aware blocks while keeping the base AR model frozen, CPD mitigrates catastrophic forgetting and offers stronger expressivity than the simple embedding concatenation used in $\sigma$-GPTs.

---

> ### Author Response · Authors · 2025-11-21
> **Response to Reviewer nPtM (PART II)**
>
> > (W1) The paper claims "teacher-forcing... is a problem in NTP" (lines 68-69) and defines a standard MLE objective in Eq. 2, but it never clearly explains why teacher-forcing is the problem it aims to solve, nor how its method (which also appears to be a supervised objective) avoids this supposed problem.
>
> 1. **Why teacher-forcing is the problem it aims to solve**：
>
>     - We would like to clarify that the actual problem we aim to address is **the lack of permutation invariance** under vanilla NTP, which manifests as the model's inability to produce approximately the same joint probability distributions across various permutations of the same sample. The primary reason is that the model is trained strictly with a left-to-right training paradigm, which inherently maximizes the likelihood only for the permutations seen during training. This limitation gives rise to the issues of reversal curse, factorization curse, and knowledge position sensitivity. We sincerely appreciate the reviewer’s concern, and we have revised the manuscript to address this point and avoid any misunderstanding.
>     - **Why permutation invariance can address the three failure modes**: Taking the reverse curse as an example, consider the sentence "*Paris is the capital of France*". Under a strict left-to-right paradigm, the model is trained to maximize P(*France*∣*Paris is the capital of*) (i.e., the MLE in Eq. 2). However, if we expect the model to correctly answer "*The capital of France is*___", we need to maximize P(Paris∣The capital of France is). This can be achieved by left-to-right training on the permuted sentence "*The capital of France is Paris*" . This illustrates that the model must learn from multiple permutations of the same sample to handle scenarios that require permutation invariance. Furthermore, Figure 1 shows the joint probability distributions of models trained with NTP, TPM, and CPC under different permutations. Simply permuting the data (as in TPM) does not guarantee similar joint probability distributions across arbitrary permutations. Therefore, our goal is to develop a method that enables mainstream AR-based LLMs to achieve better permutation invariance.
> 2. **How our method avoids this supposed problem:**  To enable the model to produce consistent joint probability distributions across arbitrary permutation patterns, we first analyze the reasons why TPM fails. The main causes are **supervised label conflicts**. To address this, we propose a position-aware modeling training framework that introduces target position-aware embeddings. By incorporating these embeddings, the model learns to condition its predictions not only on the preceding content but also on the target position in the original sequence, thereby mitigating conflicting supervision signals of token permutations. Then, we design two strategies: content–target position coupling and decoupling. This framework imparts permutation invariance to the model under vanilla left-to-right NTP. The consistent joint probability distributions observed across different permutations in Figure 1 provide empirical evidence of the effectiveness of our approach.

---

> ### Author Response · Authors · 2025-11-21
> **Response to Reviewer nPtM (PART III)**
>
> > (W2) - The authors use highly informal language, such as the "downside of potentially messing with the original token embedding" (line 109), which is inappropriate for a scientific paper.
> >
> > (W3) The paper introduces concepts like "semantic representations learned during pre-training to drift" (lines 269-271) without providing a clear definition of what this "drift" is, why it's a problem, or how the proposed 'CPD' mechanism resolves it.
>
> We thank the reviewer for pointing out the imprecise language. We acknowledge that our original phrasing was informal and the technical motivation for CPD was not sufficiently clear. We revise these sections to provide a rigorous explanation.
>
> 1. The original phrase  `downside of potentially messing with the original token embedding`  (line 109) has been revised to  `its direct integration of target position-aware embedding and input embeddings may degrade the capabilities acquired during pre-training`. This change is intended to emphasize that introducing target positional information into the original token embeddings at the input layer may interfere with the abilities the model has already acquired, which constitutes the primary motivation for proposing CPD. We appreciate your valuable suggestion and have accordingly revised the entire manuscript to replace informal expressions.
> 2. Regarding the original phrase "semantic representations learned during pre-training to drift" (lines 269–271), we have further elaborated that its intended meaning refers to the gap introduced between the pre-training and fine-tuning stages due to the explicit incorporation of target positional information at the input layer. This gap may lead to a degradation of the model's previously acquired capabilities, as discussed in detail in Appendix E.4. The overall revision is as follows:
>
>     `While CPC requires minimal modifications in the pre-trained AR model's input layer, its direct coupling of content (input embedding) and target position information (target position-aware embedding) introduces the potential semantic drift. Specifically, during pre-training, the model primarily learns to predict tokens based on their preceding content. When position-aware embeddings are directly integrated into the input representation during permutation training, the content representations learned during pre-training are modified, which could degrade the model's acquired abilities. `
>
>     Additionally, **semantic drift** can be attributed to CPC’s direct coupling of content (i.e., embeddings of the input sequence) and position (i.e., target position-aware embeddings). Therefore, we propose employing cross-attention within the position-aware block to decouple content and position explicitly: target position-aware embeddings serve as the query to guide next-token prediction, while content embeddings act as the key and value to provide the necessary information for predicting the next token.
>
>
> > (W4) Equation 3 is referred to as an "objective" (line 172), but it does not appear to be an objective function to be optimized.
>
> We achieve this objective by maximizing the joint probability distribution over all possible permutations, thereby training the model to optimize across all permutation orderings. To enhance clarity and transparency of the training objective, **we have reorganized the mathematical formulations with improved logical coherence in the revised version**. **The optimization objective is now formally presented in Equation 4.**
>
>
> > (Q1) The paper states (lines 313-315) that the attention mask "retains [the] left-to-right generation constraint in vanilla NTP." If generation at inference time is still strictly left-to-right, what is the effect of your position-aware training? How does the model's behavior at inference time differ from a standard autoregressive model?
>
> We thank the reviewer for this question. **Yes, our method still performs strictly left-to-right standard autoregressive decoding at inference time, identical to standard autoregressive models.**
>
> The effect of position-aware training:
>
> Position-aware training fundamentally changes **what the model learns**, even though the decoding process remains strictly left-to-right:
>
> 1. **Eliminates directional bias**: The model acquires multi-directional understanding rather than relying on a left-to-right understanding pattern. For example, it can learn that "A is B’" and "B is A" express the same factual relation, thereby mitigating the reversal curse, the factorization curse, and positional biases.
> 2. **Learning permutation-invariant content representations**: Across different permutations of the same sample, the model learns knowledge independently of any specific prefix context. This enables it to learn future knowledge points within a sample without relying on preceding content and, in low-diversity settings, to learn underlying facts from multiple contexts.

---

> ### Author Response · Authors · 2025-11-21
> **Response to Reviewer nPtM (PART IV)**
>
> > (Q2) The proposed method appears to add extra parameters to the model. Is it possible that any observed performance gains are simply due to an increase in model capacity? Please provide a precise breakdown of the parameter overhead introduced by your method and include comparisons to baselines with a similar total parameter count.
>
> The performance gains are **not** simply due to increased model capacity. The additional parameters in CPD serve a specific purpose: **decoupling content and position representations through additional position-aware blocks**, endowing the model with permutation invariance.
>
> **Parameter Overhead Analysis:**
>
> We provide a comprehensive breakdown in **Section 4.4 and Appendix E.5.1**:
>
> - **For Llama-3.2-1B,**  CPD-6L introduces 0.63B additional parameters with 73% training overhead and 21% inference overhead, but **achieves 63.0% EM versus NTP's 0% EM** on the reversal task.
> - **For Llama-2-7B,**  CPD-6L introduces 1.18B additional parameters and **achieves 48.3% EM versus 0% baseline**.
> - **Llama-3.2-1B-Base+CPD-6L (1.86B) significantly outperforms the much larger Llama-2-7B baseline (6.74B) by 63.0% vs 0% EM**, despite having only 28% of its parameters, demonstrating that performance gains stem from position-aware architectural design rather than increased capacity.
> - **Comparison on same base model**: To isolate architectural effects from capacity increases, we conducted a controlled experiment by expanding Llama-3.2-1B-Base from 16 to 27 layers, creating a deeper baseline with nearly identical parameter count to CPD-6L. Specifically, we duplicated the final pretrained transformer layer 11 times (i.e., copying the 16th layer's pretrained weights), resulting in a 27-layer model with 1.90B parameters (compared to CPD-6L's 1.86B). This baseline was trained under the same hyperparameters as CPD, with the only difference being the architectural design. The experimental results are shown below:
>
>   | **Llama-3.2-1B-Base**                | #P    | N2D in N2D     | N2D in D2N     |
>   | ----------------- | ------- | ---------------- | ---------------- |
>   |                 |       | EM/R-1/BLEU    | EM/R-1/BLEU    |
>   | +NTP (27 Layer) | 1.90B | 70.3/72.7/74.9 | 0.3/3.0/6.6    |
>   | +TPM (27 Layer) | 1.90B | 66.3/69.3/72.3 | 33.0/61.2/65.8 |
>   | +CPD-6L         | 1.86B | 81.3/94.7/95.8 | 63.0/85.3/87.7 |
>
>   **Analysis:**  Despite a nearly approximate parameter (\~1.9B), CPD-6L achieves  **+11.0 EM and +62.7 EM absolute improvements** over the 27-layer NTP baseline on forward and reverse tasks, respectively, and  **+30.0 EM improvement over the 27-layer TPM** on the challenging reverse task,  shows that simply adding more layers and parameters cannot replace position-aware training.
>
> > (Q3) The authors use the term "permutation-invariant language modeling" (line 485). Language is inherently sequential and not permutation-invariant (e.g., "dog bites man" vs. "man bites dog"). What is meant by "permutation-invariance" in this context?
>
> We thank the reviewer for raising this concern. We would like to clarify that **permutation-invariance**, in our work, is defined on the premise of preserving semantic dependencies.. The **goal** is for the model to **produce approximately consistent joint probability distributions across different permutations of the same sample**. In other words, permutation invariance does NOT imply **permuting the semantic order of the sequence**, i.e., "dog bites man" is not equivalent to "man bites dog".
>
> Specifically, for the original natural language sequence "dog bites man", the corresponding tokens are `["dog", "bites", "man"]` with position ids `[0, 1, 2]`. For a permuted sample of the original sequence, where the tokens appear as `["man", "bites", "dog"]`, the position ids are `[2, 1, 0]`, **NOT** `[0, 1, 2]`. Since Transformers are inherently position-agnostic, positional information is conveyed through positional encodings. By default, positional encodings assign indices `[0, 1, 2]` to the sequence. **However, we can modify these indices (position_ids) to reflect the true semantic positions.**
>
> **The key distinction between position_ids**   `[0, 1, 2]`  **vs.**  `[2, 1, 0]`:
>
> - **Position ids** `[0, 1, 2]`: This would represent a completely new sequence with entirely different semantics, i.e., "man bites dog".
> - **Position ids** `[2, 1, 0]`: The model predicts the token at position 1 ("bites") conditioned on the token at original position 2 ("man"); then predicts the token at position 0 ("dog") conditioned on tokens at original positions 2 and 1 ("man" and "bites"). This preserves the semantic meaning "dog bites man".

---

### Official Review · Reviewer_Qq6w · 2025-10-30

**Soundness:** 2
**Presentation:** 1
**Contribution:** 2
**Rating:** 2
**Confidence:** 4

**Summary:**

LLM's are mainly trained in a teacher forced next token prediction setting. As a result the model mainly captures the entangled training distribution, and will not appropriately handle scenarios like token permutation, as they'd be required to be added to training datasets.
The authors therefore propose training some additional parameters to the attention mechanics, in order to disentangle token order.

**Strengths:**

- it is in principle a relevant field of research, e.g. making LLM's more reusable in different scenarios like those permutation settings

**Weaknesses:**

W1 the paper is unfortunately in its current form poorly written which already is not keeping the bar for this conference. there are plenty grammar issues and weird phrases in abstract and introduction. the figure 1 does not help at all (CPC looks like plain transformer) and the pseudo code is also not really presentable.

W2 the permutation curse is only partially covered. yes, we have token permutation now, but this almost never exist in the language case, except for permutation of whole sentences, for which other approaches seem more plausible. (E.g. a reversed written word will end up in completely different sequence of tokens. if it was discussed i missed it, but it should be there!)  The TPM experiments , e.g. what kind of permutation was used in main text, are hard to follow, i.e. if they are comparable at all to what you do in CPC. Yes on graph or number shuffling it makes more sense, here it's unclear to me however, if its 'just your embedding technique' or if you trained the algorithm into that added attention mechanics.

W3 you want 'minor modifications' - yet you afterall train positional embeddings and another crossattention which most likely renders the model useless on plain language when 'running in this mode'. if quick reusability/ exchangeability of deployed models is your goal i'd require an ablation against adapters etc.

**Questions:**

none

---

> ### Author Response · Authors · 2025-11-21
> **Response to Reviewer Qq6w (PART I)**
>
> Thank you very much for taking the time to review and for your support. We try our best to address your questions as follows.
>
> > W1 the paper is unfortunately in its current form poorly written which already is not keeping the bar for this conference. there are plenty grammar issues and weird phrases in abstract and introduction. the figure 1 does not help at all (CPC looks like plain transformer) and the pseudo code is also not really presentable.
>
> Thank you for your feedback. We address each concern:
>
> 1. **Grammar and Language Quality.**  We conducted multiple rounds of careful proofreading before submission to ensure academic rigor. Regarding "**weird phrases**", we intentionally employed formal academic language that may differ from colloquial usage, which may have led to some misunderstandings. **We have made our best efforts to revise them in the revised version.**
> 2. **Figure 1 Clarity (Perhaps actually referring to Figure 2)** .
>
>     Our proposed CPC **does not modify the plain transformer architecture**. Instead, **it operates at the input layer** by introducing target position-aware embeddings and integrating them into the input token embeddings (**as we claim, CPC is a minimal modification for AR-based LLMs**).
>
>     However, **we apologize, but we are unable to agree with your statement that "Figure 1 does not help at all",**  as this figure serves dual purposes:
>
>     - **Left (CPC)** : Clearly illustrates how target position-aware embeddings are **coupled** with input embeddings via element-wise addition, representing a minimal-modification approach.
>     - **Right (CPD)** : Demonstrates how target position-aware embeddings serve as **Queries** in the cross-attention layer of the position-aware block, with base transformer hidden states as **Keys/Values,**  achieving explicit **decoupling** of content and position.
>
>     **Could you specify what additional information would make this figure more helpful?**  We are committed to improving clarity but need concrete guidance on what is missing in your view.
> 3. **Pseudo-Code Presentation. Our Python-style pseudo-code prioritizes:**
>
>     - **Reproducibility**: Clear implementation details help with reproduction.
>     - **Transparency**: Shows tensor shape transformations at each step.
>     - **Technical precision**: Matches actual code structure.
>
>     This style is common in academic papers for algorithmic contributions.

---

> ### Author Response · Authors · 2025-11-21
> **Response to Reviewer Qq6w (PART II)**
>
> > W2 the permutation curse is only partially covered. yes, we have token permutation now, but this almost never exist in the language case, except for permutation of whole sentences, for which other approaches seem more plausible. (E.g. a reversed written word will end up in completely different sequence of tokens. if it was discussed i missed it, but it should be there!) The TPM experiments , e.g. what kind of permutation was used in main text, are hard to follow, i.e. if they are comparable at all to what you do in CPC. Yes on graph or number shuffling it makes more sense, here it's unclear to me however, if its 'just your embedding technique' or if you trained the algorithm into that added attention mechanics.
>
> Thank you for raising this important concern. We would like to clarify several points:
>
> 1. **Regarding permutation applicability**: Both sentence-level and token-level permutations have distinct practical scenarios. Sentence permutation is suitable when each sentence contains isolated knowledge points (e.g., positional bias mitigation in Wiki2023+). However, token permutation is essential for intra-sentence knowledge reversal, such as in the reversal curse: during training, the model sees "Paris → France" in "Paris is the capital of France," but at test time encounters "France → Paris" in "The capital of France is \_\_\_." This requires token permutation within a single sentence.
> 2. **Regarding permutation granularity**: **Token permutation operates at the word level (multiple tokens forming words), not individual sub-tokens.**  Specifically, for the phrase `"understanding token permutation"`, the tokenizer produces `['under', 'standing', 'Ġtoken', 'Ġpermutation']`. Except for the first token, we treat tokens beginning with the special start marker `"Ġ"` as standalone words during training, and this is handled dynamically during training. Therefore, we do not permute sub-tokens within a word. Instead, we treat the whole word or n-words as permutation units. This avoids the issue you mentioned, i.e., a reversed written word like `"standingunder"` never occurs. Table D2 in the Appendix provides concrete examples illustrating permutation granularity. It is worth noting that although CPC trains on permuted sequences, we explicitly retain each token's original position information through the `position_ids` parameter passed to the transformer's inherent positional encoding mechanism. This means we only adjust the *order* in which content is learned, not the content itself.
> 3. **Regarding TPM experimental setting**: We provide detailed TPM implementation specifications in Appendix D.2.4, D.3.4, and D.4.4, including permutation strategies. **CPC and TPM use identical permutation approaches**, making them directly comparable. The only difference lies in whether target position-aware embeddings are incorporated.
> 4. **Regarding the question about "embedding technique vs. attention mechanics"** : the position-aware embedding is indeed the foundational component shared by both CPC and CPD. CPC directly adds this embedding to the input (content) embedding, while CPD introduces it through cross-attention mechanisms. Therefore, it is the **position-aware modeling under permutation** that drives performance gains, while the additional attention mechanism in CPD serves specifically to *decouple* position from content, thereby preserving the base LLM's general capabilities (as demonstrated in Appendix E.4).

---

> ### Author Response · Authors · 2025-11-21
> **Response to Reviewer Qq6w (PART III)**
>
> > W3 you want 'minor modifications' - yet you afterall train positional embeddings and another crossattention which most likely renders the model useless on plain language when 'running in this mode'. if quick reusability/ exchangeability of deployed models is your goal i'd require an ablation against adapters etc.
>
> Thank you for this important concern. We clarify our design and provide experimental evidence on plain language tasks:
>
> 1. **Regarding "minor modifications"** : We specifically characterize **CPC** as requiring minor modifications because it only trains a single additional target position-aware embedding without modifying the original architecture or introducing architectural components like the cross-attention layers in CPD. This minor modification aligns with the goal of rapid reusability and exchangeability for deployed models in tasks that require robustness to token order, as you correctly noted.
> 2. **Regarding "useless on plain language"** : We acknowledge this limitation and have conducted comprehensive studies (Appendix E.4) evaluating both CPC and CPD on standard language benchmarks. The results confirm your concern: when the position-aware embedding is removed during inference, **CPC indeed suffers from performance degradation on plain language tasks due to the training-inference paradigm mismatch**. This is precisely why we introduced **CPD** as a complementary approach. In **plain language tasks,**  the position-aware block can be removed without affecting the base LLMs' inherent abilities. Our experiments in Appendix E.4 (specifically Table E3 and E4) demonstrate that CPD achieves an effective balance between mitigating NTP failure modes and maintaining performance on plain language tasks.
> 3. **Regarding ablation against adapters**: Our CPD design inherently follows adapter principles. The experiments in Appendix E.4 effectively serve as adapter-style ablations, systematically varying which components are trainable (embedding only, specific transformer layers, position-aware blocks). Table E4 demonstrates that selectively fine-tuning upper layers preserves general capabilities while enabling position-aware reasoning. We make this connection to adapter methodologies more explicit in the revision to clarify CPD's modular, plug-and-play nature.

---

### Official Review · Reviewer_trsa · 2025-10-31

**Soundness:** 2
**Presentation:** 2
**Contribution:** 3
**Rating:** 2
**Confidence:** 4

**Summary:**

This paper argues that current next-token prediction training paradigm could be improved to the proposed content-position coupling and content-position decoupling to improve modeling for positional informations. The key is to incorporate a ROPE embedding (as in equation (1)) into the next-token prediction training. Content-position coupling directly injects the rope embedding into the input IDs, while content-position decoupling uses an attention module to incorporate that. Through experiments on the reversal curse, factorization curse, and positional bias, this paper demonstrates the improvement of their methods over standard NTP.

**Strengths:**

The research question is intuitive on current failure modes of NTP training. The formulation is rigorous and the experiment results looks good to me. The code is provided for reproducibility.

**Weaknesses:**

1. To be honest I still don't quite get why the core idea is working. Why is incorporating z_{τ_t} from equation (1) improving the current transformer architecture that is already considering positional embeddings such as RoPE? Is this work trying to make the RoPE positional embedding more explicit for NTP?

2. Following 1, why does the proposed method improve performance on the reversal curse and factorization curse, which seems far from positional embeddings to me? In summary, I don't quite get the intuition why the proposed CPC and CPD work.

3. I was looking for experiments on natural language modeling, but maybe I missed it. The experiment on Wiki2023+ is only about movie domain according to line 428. It is uncertain to me if CPD would harm NLP/reasoning performance and experiments are needed here. Optionally, the authors could explicitly note in this paper that though CPC/CPD gives SOTA performance on path planning, alg reasoning, etc., it is not tested yet if this method could actually be used to pretrain language models.

4. When reading CPD, I was thinking about the additional attention cost (approximately double?) when letting CPD to run additional attention with the same length of sample. The cost for CPC should be minimal as it directly incorporate the RoPE embedding. Then, I found Table E1 in appendix, which is actually important and should be extended and move to the main pages (the main pages emphasize improvement but not cost).

On Table E1, first of all, I want to ask why CPC is not a minimal improvement over NTP but TPM? Is it because we need to generate many permuted samples? In that case, the novelty of CPC is weakened by the existence of TPM. Second, there should be some study comparing marginal cost & marginal performance gain of CPD for future reference whether CPD should be implemented or not in practice. (when performance of NTP is acceptable, there is no need to implement CPD for double cost).

5. The evaluations are all on LLaMA models, which implements RoPE already to the best of my knowledge. It is uncertain if CPC/CPD are compatible with other positional encodings and other groups of LLMs.

This paper is a clear rejection at its current stage, as I was unable to grasp the core intuition behind why the proposed method works (which may partly be due to my own limitations). Given this uncertainty, it would be risky for me, a cautious reviewer, to recommend acceptance at this point. That said, the overall quality of the work appears promising, and several missing experiments could substantially strengthen the paper. I would be glad to revisit my evaluation and potentially raise my rating accordingly, even to acceptance during the rebuttal phase.

**Questions:**

1. Could the authors illustrate more about how z_τt is different from Pos_Embed(τj) in equation (6)?

2. In what scenarios do you think we should use CPC/CPD rather than NTP? (given that NTP is working pretty well now for training production LLMs)

3. Is CPD using 1-layer attention or stacked layers of attention?

---

> ### Author Response · Authors · 2025-11-21
> **Response to Reviewer trsa (PART I)**
>
> Thank you very much for taking the time to review and for your support. We try our best to address your questions as follows.
> > (W1) To be honest, I still don't quite get why the core idea is working. Why is incorporating z\_{τ\_t} from equation (1) improving the current transformer architecture that is already considering positional embeddings such as RoPE? Is this work trying to make the RoPE positional embedding more explicit for NTP?
> >
> > (Q1) Could the authors illustrate more about how z\_τt is different from Pos\_Embed(τj) in equation (6)?
>
> We thank the reviewer for raising this core question. We clarify as follows:
>
> - $\boldsymbol{z_{τ_t}}$ **vs. positional encodings in current transformer architecture:**   Existing positional encodings in transformer architectures (e.g., RoPE) model the position of the current token in the sequence, whereas $z_{τ_t}$ model the target token's  (i.e., the next token to be predicted) position in the original sequence. Specifically:
>
>   Consider the original sequence: `["<bos>", "The", "cat", "sat", "on", "mat", "<eos>"]` with position indices `[0, 1, 2, 3, 4, 5, 6]`. After permutation: `["<bos>", "sat", "on", "mat", "The", "cat", "<eos>"]` with position indices `[0, 3, 4, 5, 1, 2, 6]`.
>
>   Taking CPC as an example, when predicting "sat" in the permuted sequence, it uses $\phi(\text{Embed("\<bos\>")}\oplus z_3, \text{Pos}$_$\text{Embed(0)})$:
>
>   - $\phi(\cdot,\cdot)$ is a token-position fusion function, $\oplus$ denotes the interaction operation between content and target position (element-wise addition as the default setting).
>   - **Pos_Embed(0):**  encodes the position of the input token "\<bos\>" in the current permuted sequence (position 0)
>   - $z_3 (z_{\tau_1})$: encodes the position of the target token "sat" in the original sequence ($τ_1=3$)
>
>   (**Respond to Q1**) **Pos_Embed()**  represents the position of the current token in the sequence, while $z_{\tau_{t}}$ represents the target token's position in the original sequence.
> - **Why the core idea is working:**   see response to W2.
>
>
> > (W2) Following 1, why does the proposed method improve performance on the reversal curse and factorization curse, which seems far from positional embeddings to me? In summary, I don't quite get the intuition why the proposed CPC and CPD work.
>
> We thank the reviewer for this insightful question. We clarify below how CPC and CPD are connected to the observed improvements on the reversal curse and factorization curse.
>
> The reversal curse and factorization curse mainly because **the model lacks permutation invariance**, i.e., the inability to produce approximately consistent joint probability distributions across different permutations of the same sample (Eq. 3). The most direct solution is to train models on many permuted samples (e.g., Token Permutation, TPM), maximizing the sum of log-probability under as many permutations as possible (Eq. 2) to approximate the objective in Eq. 4. However, TPM introduces a critical challenge under permutation training through vanilla NTP: **the model fails to converge  (as evidenced in Figures E2 and E3). This failure arises for two reasons:**
>
> 1. **Data-distributional mismatch**: The model treats the permuted sentences as if they were in a normal order, without recognizing that they have been permuted. This may not pose a problem when training a model from scratch. However, when a model pretrained on normal order is continued training on permuted sequences, the coexistence of normal and permuted sequences creates a substantial distributional mismatch.
> 2. **Supervision signal conflict:**  **The same prefix may correspond to different suffixes** (detailed in Appendix E.1). Specifically, when the model encounters the same prefix in different permutations but predicts different next tokens, it cannot distinguish which token to predict without explicit target position information.
>
> **Our Solution:**
>
> - **Target position-aware** **embedding:**  explicitly model the position of the next token to be predicted. By providing this explicit target position signal, the model can
>
>   - distinguish between normal and permuted sequences.
>   - distinguish cases where identical prefixes require different predictions based on the target position.
> - **CPC and CPD:**
>
>   - Content-Position Coupling (CPC): directly adds the target position-aware embedding to the content embeddings (input sequence embeddings).
>   - Content-Position Decoupling (CPD): We further observed that when CPC is directly applied to pretrained LLMs, it can affect the model's basic capabilities. We thus introduce modular position-aware blocks with incremental parameters, where the target position-aware embedding serves as the query and the content embedding as the key and value. This design minimally affects the original model's capabilities while preserving permutation invariance (see detailed experiments in Appendix E.4).

---

> ### Author Response · Authors · 2025-11-21
> **Response to Reviewer trsa (PART II)**
>
> > (W3) I was looking for experiments on natural language modeling, but maybe I missed it. The experiment on Wiki2023+ is only about movie domain according to line 428. It is uncertain to me if CPD would harm NLP/reasoning performance and experiments are needed here. Optionally, the authors could explicitly note in this paper that though CPC/CPD gives SOTA performance on path planning, alg reasoning, etc., it is not tested yet if this method could actually be used to pretrain language models.
>
> We thank the reviewer for this important concern and clarify our experiments as follows:
>
> 1. **Impact on NLP/Reasoning Performance:**  We share the reviewer's concern about potential negative impacts on general capabilities. We provided comprehensive evaluations in **Appendix E.4** on **nine standard NLP/Reasoning benchmarks**: BoolQ, PIQA, SIQA, HellaSwag, WinoGrande, ARC-easy, ARC-challenge, OpenBookQA, and 5-shot MMLU, demonstrating that **CPD achieves a balance between mitigating NTP failure modes and preserving original capabilities.**
> 2. **Clarification on Pretraining Scope:**  As noted in our Limitations section (Appendix A), our work focuses on continued pretraining and fine-tuning rather than full-scale pretraining from scratch due to the huge cost.
>
>
>
> > (W4) Regarding the location of the experiment on performance and computational consumption.
>
> We sincerely thank the reviewer for this valuable suggestion. We have moved the appendix content to section 4.4 in the main pages (Lines 451-475).
>
>
> > (W4) On Table E1, first of all, I want to ask why CPC is not a minimal improvement over NTP but TPM? Is it because we need to generate many permuted samples? In that case, the novelty of CPC is weakened by the existence of TPM. Second, there should be some study comparing marginal cost & marginal performance gain of CPD for future reference whether CPD should be implemented or not in practice. (when performance of NTP is acceptable, there is no need to implement CPD for double cost).
>
> We thank the reviewer for these insightful questions. We clarify as follows:
>
> 1. CPC vs. TPM
>
>     - **CPC is indeed built upon TPM, but addresses its fundamental flaw:**  As discussed in **W2**, TPM has difficulties in converging when approximating Eq. 4, mainly due to two reasons: **data-distributional mismatch** and **supervision signal conflict**.
>     - CPC's Novelty: CPC addresses both weaknesses through **lightweight position-aware embeddings**:
>
>       - **Addressing data-distributional mismatch**: By explicitly encoding the target token's position in the original sequence, CPC provides a bridge between the permuted and the original sequence, mitigating the distributional mismatch.
>       - **Addressing supervision signal conflict**: Target position-aware embeddings enable the model to explicitly distinguish "**what to predict**" (content) and "**where to predict**" (target position), eliminating the supervision ambiguity.
> 2. Marginal Cost-Performance Analysis of CPD
>
>     We appreciate this suggestion. Following the reviewer's guidance, we provide an **extended cost-performance analysis** complementing Table E5. Results on Llama-2-7B and Llama-3.2-1B (reversal curse, challenging D2N in N2D task):
>
>     |Llama-2-7B|#P (B)|$\Delta$#P %|EM|
>     |-|-|-|-|
>     |NTP|6.74|0|0|
>     |CPD-8L|8.25|22.40|47.6|
>     |CPD-12L|8.93|32.49|52.6|
>     |CPD-24L|10.97|62.76|59.7|
>     |CPD-30L|12.15|80.26|59.3|
>     |CPD-35L|13.01|93.03|53.3|
>
>     |Llama-3.1-1B|#P (B)| $\Delta$#P %|EM|
>     |-| - | - | - |
>     | NTP| 1.23| 0 | 0|
>     | CPD-1L| 1.57| 27.64 | 31.5 |
>     | CPD-3L| 1.68| 36.59 | 43.7 |
>     | CPD-6L| 1.86| 51.21 | 63.0 |
>     | CPD-8L| 1.98| 60.98 | 63.4 |
>     | CPD-12L| 2.21| 79.67 | 65.8 |
>     | CPD-14L| 2.32| 88.62 | 61.5 |
>
>     **Key Observations:**
>
>     1. **Relative** **Parameter Threshold:**  Without carefully tuning hyperparameters, we observe that **CPD requires at least a** **20-30% relative parameter increase** to achieve substantial performance gains. Below this threshold (e.g., CPD-1L for Llama-2-7B), the additional parameter is insufficient to effectively learn permutation-invariance.
>     2. **Optimal Performance Range (60-80% relative parameter increase):**  For both Llama-2-7B and Llama-3.2-1B, performance improves steadily as CPD is expanded by a 20–60% relative increase in parameters, reaches its best performance when the relative increase is between 60–80%, and then either stably fluctuates or slightly declines once the increase exceeds 80%.
>
>     **Practical Recommendations for Deploying CPD:**
>
>     1. **Maximum Performance Priority**: Allocate a 60–80% relative increase in parameters for CPD.
>     2. **Parameter Efficiency Priority**: Allocate a 20–30% relative increase in parameters for CPD.
>     3. **Minimal Budget Scenario**: When the CPD-specific relative parameter increase is below 20%, the performance gains are minimal compared with CPC. **In such cases, consider adopting CPC, which requires nearly 0% additional parameters.**

---

> ### Author Response · Authors · 2025-11-21
> **Response to Reviewer trsa (PART III)**
>
> > (W5) The evaluations are all on LLaMA models, which implements RoPE already to the best of my knowledge. It is uncertain if CPC/CPD are compatible with other positional encodings and other groups of LLMs.
>
> We thank the reviewer for this important question regarding generalizability. We clarify as follows:
>
> The **core principle of CPC/CPD, modeling the relationship between content and target position,**  is independent of the specific positional encoding used by the base LLMs. We emphasize this important distinction:
>
> 1. **CPC's Compatibility:**  CPC introduces an **additional target position-aware embedding** that is conceptually and architecturally separate from the base LLMs' existing positional embedding:
>
>     - **Base LLMs' positional encoding** (e.g., RoPE, Learned Absolute Position Encoding, etc): Encodes the position of current tokens in the sequence.
>     - **CPC's target position-aware embedding** ($z_{\tau_t}$): Encodes the position of the next token to be predicted in the sequence.
> 2. **CPD's Independence:**  CPD is even more independent from the base LLMs' positional encoding. CPD appends modular position-aware blocks on top of the base LLMs' final layer.
>
> To validate our claim of architecture independence, we conducted experiments on **GPT-2-LARGE, which uses learned absolute position encodings** instead of RoPE. Results on the reversal curse (name-description dataset):
>
> | Method | N2D in N2D     | D2N in N2D        | N2D in D2N     | D2N in D2N        |
> | -------- | ---------------- | ------------------- | ---------------- | ------------------- |
> |        | EM/R-1/BLEU    | EM/R-1/BLEU       | EM/R-1/BLEU    | EM/R-1/BLEU       |
> | NTP    | 83.7/85.1/86.7 | 0.0/0.0/0.1       | 0.7/2.3/7.3    | 100.0/100.0/100.0 |
> | TPM    | 57.3/61.0/64.3 | 91.7/91.7/91.7    | 37.0/47.7/52.4 | 100.0/100.0/100.0 |
> | CPC    | 68.3/71.5/75.0 | 100.0/100.0/100.0 | 54.3/62.0/65.9 | 100.0/100.0/100.0 |
> | CPD-6L | 69.0/72.7/76.5 | 100.0/100.0/100.0 | 58.7/67.7/71.5 | 100.0/100.0/100.0 |
>
> Both CPC and CPD mitigate the reversal curse on GPT-2, achieving better performance than TPM.
>
>
>
> > (Q2) In what scenarios do you think we should use CPC/CPD rather than NTP? (given that NTP is working pretty well now for training production LLMs)
>
> While NTP performs well in large-scale pretraining (primarily due to massive, diverse corpora providing implicit coverage of different token orderings), CPC/CPD become valuable in scenarios where NTP's inherent limitations are exposed.
>
> **Use CPC/CPD when:**
>
> 1. Resource is constrained:
>
>     - Limited access to large-scale diverse training data: knowledge appears in limited variations (e.g., domain-specific, emerging events, specialized corpora, and etc).
>     - Need to utilize smaller models for efficiency in limited data diversity.
> 2. Permutation-invariation ability is required: Training data contains "A→B" but inference requires "B→A" (reversal curse) or alternative factorizations (factorization curse).
>
> **Use** **NTP when:**
>
> 1. Large-scale pretraining with naturally diverse corpora where implicit coverage of orderings is adequate.
> 2. Specific tasks where token order doesn't harm performance.
>
> > (Q3) Is CPD using 1-layer attention or stacked layers of attention?
>
> We thank the reviewer for seeking this clarification. **CPD uses stacked layers of cross-attention.**  Specifically, CPD appends **M position-aware blocks** after the base LLMs' final layer, where each position-aware block contains one cross-attention layer. This results in a total of **M layers of cross-attention**. We set M \= 6 by default throughout our experiments (as stated in Appendix D, line 853: "In this paper, if CPD variants are not specifically stated, our default CPD block number M \= 6 is used"). Additionally, we provided ablation experiments on the number of CPD layers in **Table E5**.

---

> > ### Comment · Reviewer_trsa · 2025-11-21
> > **Thank you for your clarifications. Please be tuned!**
> >
> > I appreciate the authors' timely response. At a glance, the point-by-point clarifications seem helpful and will likely improve my understanding of the techniques and contributions.
> >
> > I am currently occupied and cannot review the responses in depth immediately, but I will carefully read both the rebuttal and the revised paper (if available) and follow up as soon as possible. I will do my best to complete this before Sun, Nov 23.

---

> > ### Comment · Reviewer_trsa · 2025-11-25
> >
> > > W1 & Q1
> >
> > Just want to double-check, is positional encoding learns the position in the whole sequence, while this work focuses on the relative position within part of the sequence, e.g., the order in a single sentence?
> >
> > > W2
> >
> > Clear. Thanks!
> >
> > > W3
> >
> > Great to see the additional experiments on language modeling, and I’m glad the proposed method achieves a reasonable balance between language-modeling performance and robustness to the reversal curse. That said, I want to note that the reversal curse may not always be a critical concern. Many practical applications prioritize strong language-modeling ability in settings where input order is less important. With this in mind, it would be helpful for practitioners if the paper more clearly articulated the intended scope and scenarios where the proposed method is most appropriate.
> >
> > > W4
> >
> > Great and thanks for the extended marginal cost study. I'm clear on this and would recommend spending some space in the main page to cover it.
> >
> > > W5
> >
> > I appreciate the additional experiment on GPT-2 but I do want to note that this model is kind of out-dated and is relatively small. I understand that the time of rebuttal phase is limited, but I would recommend that authors test the proposed methods on more recent language models beyond the LLaMA family.
> >
> > > Q2 & Q3
> >
> > Both questions are clear to me now. Please do discuss the advice for practitioners (could be in the appendix) and make the attention layer configuration disambiguous.
> >
> > Overall, the authors’ clarifications and additional experiments helped me better understand the contribution of this work. I am increasing my score (from 2 to 4). From my perspective, the revised manuscript remains borderline: although the new additions are valuable, due to the original presentation quality, the number of substantial revisions made during the rebuttal period makes it difficult to fully assess the stability and quality of the final version. As reviewers, we also carry prior context from the original submission, which may not be shared by future readers, so it is not entirely clear how the revised paper will be perceived by those encountering it for the first time.

---

### Official Review · Reviewer_pECw · 2025-11-01

**Soundness:** 3
**Presentation:** 3
**Contribution:** 3
**Rating:** 6
**Confidence:** 4

**Summary:**

The paper identifies a single root cause for several well-known failure modes in Large Language Models (LLMs), such as the reversal curse, factorization curse, and knowledge position sensitivity. The authors argue that these issues stem from the standard Next-Token Prediction (NTP) training paradigm, where the fixed left-to-right token order entangles a token's content with its position. To solve this, they propose a position-aware training framework that explicitly disentangles these two aspects. By making the model aware of the intended position of the token it is predicting, it can learn more robust, permutation-invariant representations.
The core solution is to augment the training process by providing the model with explicit information about the target position of the token to be predicted. This is done through two complementary approaches:

1. Content-Position Coupling (CPC): This is a simple, data-centric approach. A lightweight, learnable position-aware embedding is created for the target token's position. This embedding is then directly added to the input token embeddings before they are fed into the transformer. It requires no changes to the base model's architecture. The model implicitly learns to use this fused information to distinguish between different target positions.

2. Content-Position Decoupling (CPD): This is a more modular, model-level approach. It adds auxiliary position-aware blocks on top of a pre-trained AR model. These blocks use a cross-attention mechanism where the Query is the target position embedding. The Key and Value are the hidden states from the base model. This design explicitly separates the positional query from the content, allowing for more flexible and explicit supervision.

**Strengths:**

Novelty: The paper provides a clear and unified explanation for several seemingly disparate LLM failures, tracing them back to the entanglement of content and position in NTP. Both proposed methods, CPC and CPD, are shown to be highly effective at mitigating these failures across a range of tasks, from synthetic benchmarks to real-world knowledge retrieval.
Simplicity: CPC is extremely simple, requiring no changes to the model architecture, making it easy to implement. CPD is designed as a modular add-on, allowing it to be integrated with existing pre-trained models without requiring full retraining. This is a major practical advantage.

**Weaknesses:**

Scope of Evaluation: The framework has not been tested on more complex symbolic reasoning tasks, where hierarchical structure is as important as sequential position.

**Questions:**

How does this model compare on reasoning benchmarks where causality matters?

---

> ### Comment · Area_Chair_2ccU · 2025-11-21
> **AI generated review**
>
> This review seems to be AI generated with unrelated comments to the paper and therefore will be disregarded in the review and decision process.

---

> ### Author Response · Authors · 2025-11-22
> **Response to Reviewer pECw**
>
> Thank you very much for taking the time to review and for your support. We try our best to address your questions as follows.
>
> > Scope of Evaluation: The framework has not been tested on more complex symbolic reasoning tasks, where hierarchical structure is as important as sequential position.
> >
> > How does this model compare on reasoning benchmarks where causality matters?
>
> To evaluate the effectiveness of our method on complex symbolic reasoning tasks that require hierarchical structure and causality, we conduct experiments using the `Llama-3.2-1B-base` model on the original GSM8K dataset, which involves multi-step mathematical reasoning with causal dependencies between steps.
>
> **Experimental Setup:**
>
> - All methods except "Original" and "CPD-6L+" are trained on the original 7,473 examples without data augmentation. CPD-6L+ refers to training on the GSM8K-Aug dataset[1], which uses GPT-4 for data augmentation, with a total of 385,620 training samples.
>
>   [1] Implicit chain of thought reasoning via knowledge distillation.
> - All evaluations are performed on 1,319 test examples.
> - Hyperparameters are kept consistent with our factorization curse experiments.
>
> | Method   | Mode   | Acc    |
> | ---------- | -------- | -------- |
> | Original | 8-shot | 6.53%  |
> | NTP      | 0-shot | 19.86% |
> | TPM      | 0-shot | 15.22% |
> | CPC      | 0-shot | 20.75% |
> | CPD-6L   | 0-shot | 28.63% |
> | CPD-6L+  | 0-shot | 36.77% |
>
> **Key Observations:**
>
> 1. CPD-6L achieves 8.77% absolute improvement over vanilla NTP, demonstrating that position-aware modeling enhances causal reasoning capabilities.
> 2. All position-aware methods outperform TPM **,**  indicating that explicit target position information can also help mitigate supervised label conflicts in multi-step reasoning, thereby enabling the model to capture hierarchical dependencies better.
> 3. The gain in 0-shot performance suggests that position-aware methods help models learn more robust reasoning patterns that transfer to unseen problems, rather than merely memorizing training examples.
>
> While our GSM8K results are encouraging, **systematically improving generalization in complex reasoning tasks is beyond the scope of this work**. Our primary contribution is demonstrating that explicit position-aware modeling can eliminate failures in NTP. Exploring how our framework might enhance reasoning generalization through better training paradigms (rather than relying solely on data augmentation) is an interesting direction for future work.

---

### Meta-Review · Area_Chair_YqWj · 2025-12-22

**Summary:**

This paper introduces a position-aware training method that addresses permutation-related failures in LLMs, which is achieved by separating token content from token position.

The reviewers were originally leaning towards rejecting the paper. Several concerns were raised about the lack of intuition ("why does this help if RoPE already encodes position?" which is also a question that came to my mind), limited empirical evaluations, and unclear cost/benefit of the method compared to baseline methods. One reviewer was satisfied by the clarifications and empirical results provided by the authors and raised their score to borderline accept. Other reviewers remain negative (some did not engage in a discussion unfortunately), mainly citing poor writing, confusion about what permutations are used and whether the setting is realistic for language. One reviewer also criticizes insufficient comparisons to prior autoregressive work and asks whether gains could be explained by the extra parameters of the method.

This was a lot to address for the authors who managed to discuss concerns about unclear intuition, lack of reasoning/NLP evaluations, computational cost and many other comments.
However, I think that several concerns remain (and would have remain even if the reviewers could have engaged in a longer discussion with the authors), including: overall presentation quality, limited comparison to alternative autoregressive baselines, and whether the added complexity is justified for practical large-scale pretraining.

Overall, the idea is viewed as interesting and potentially impactful, but several reviewers doubt the clarity and evaluation completeness. I think these concerns are quite important and do require a significant revision of the paper. I therefore do not recommend acceptance and strongly encourage the authors to make the changes requested by the reviewers.

**Reviewer Concerns:**

See the rebuttal above where I addressed this question.

**Reviewer Scores:**

Only one reviewer increased their scores. I think some important concerns remain, including the quality of the writing and the evaluation completeness (e.g. lack of comparison to baselines).

---

### Decision · Program_Chairs · 2026-01-26

Reject